# Mcm10 promotes rapid isomerization of CMG-DNA for replisome bypass of lagging strand DNA blocks

Lance D Langston[1,2], Ryan Mayle[1,2], Grant D Schauer[1], Olga Yurieva[1,2], Daniel Zhang[1], Nina Y Yao[1], Roxana E Georgescu[1,2], Mike E O'Donnell[1,2]*

[1]The Rockefeller University, New York, United States; [2]Howard Hughes Medical Institute, New York, United States

**Abstract** Replicative helicases in all cell types are hexameric rings that unwind DNA by steric exclusion in which the helicase encircles the tracking strand only and excludes the other strand from the ring. This mode of translocation allows helicases to bypass blocks on the strand that is excluded from the central channel. Unlike other replicative helicases, eukaryotic CMG helicase partially encircles duplex DNA at a forked junction and is stopped by a block on the non-tracking (lagging) strand. This report demonstrates that Mcm10, an essential replication protein unique to eukaryotes, binds CMG and greatly stimulates its helicase activity in vitro. Most significantly, Mcm10 enables CMG and the replisome to bypass blocks on the non-tracking DNA strand. We demonstrate that bypass occurs without displacement of the blocks and therefore Mcm10 must isomerize the CMG-DNA complex to achieve the bypass function.
DOI: https://doi.org/10.7554/eLife.29118.001

*For correspondence:
odonnel@mail.rockefeller.edu

**Competing interests:** The authors declare that no competing interests exist.

## Introduction

The replication of cellular DNA requires use of a helicase to separate the interwound strands. The replicative helicase in all domains of life is a circular hexamer. There are four superfamilies of hexameric helicases, SF3-6, that assort into two main groups, the bacterial helicases (SF4,5) that have ATP sites derived from the RecA fold and the eukaryotic/archaeal helicases (SF3,6) that have ATP sites derived from the AAA+ fold (*Singleton et al., 2007*). In all cases, the subunits of hexameric helicases are composed of at least two major domains, giving them the appearance of two stacked rings, an N-tier ring and C-tier ring; the motors are contained in the C-tier ring. The RecA based bacterial helicases track 5′−3′ on DNA with the C-tier motors leading the N-tier, as determined by crystal structures of bacterial Rho (SF5) and DnaB (SF4) complexed with ssDNA, while structures of eukaryotic bovine papilloma virus E1 (SF3) and S. cerevisiae CMG (SF6) with DNA show that these helicases track 3′−5′ with the N-tier ahead of the C-tier (*Enemark and Joshua-Tor, 2006*; *Georgescu et al., 2017*; *Itsathitphaisarn et al., 2012*; *Thomsen and Berger, 2009*).

Hexameric helicases are thought to act by encircling only one strand of DNA upon which they track and exclude the non-tracking strand to the outside of the ring, thereby acting as a wedge to split DNA in a process often referred to as steric exclusion and illustrated in *Figure 1A* (*Bell and Labib, 2016*; *Enemark and Joshua-Tor, 2008*; *Lyubimov et al., 2011*). Whether a helicase functions by steric exclusion is determined by biochemical experiments that place a bulky block on one or the other strand of the duplex. For a steric exclusion helicase, a block placed on the non-tracking strand (i.e. the strand that is excluded from the central channel) does not inhibit helicase unwinding, while a block placed on the tracking strand stops the helicase because the bulky block cannot fit through the central channel.

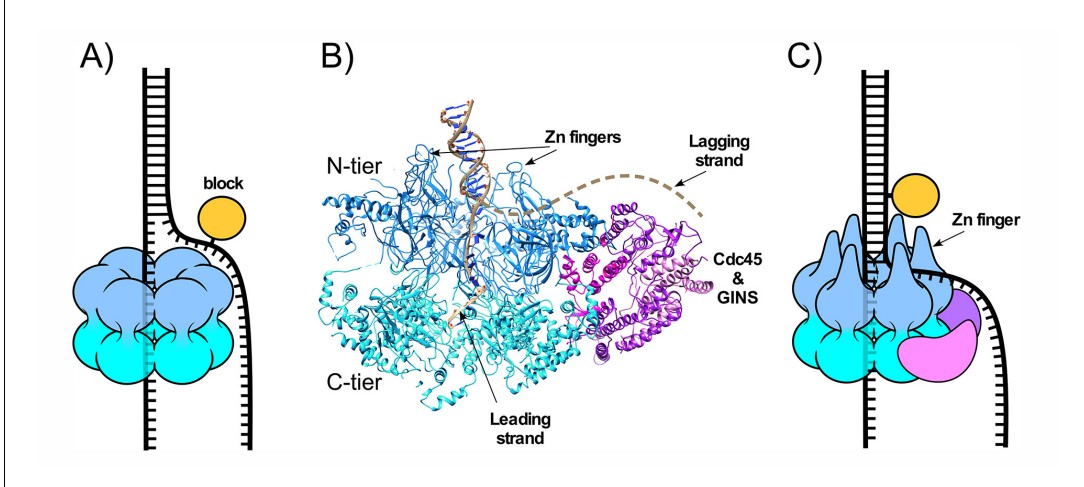

**Figure 1.** CMG-fork structure and isomerization needed for bypass of lagging strand blocks. (**A**) Illustration of a hexameric helicase in the steric exclusion mode, encircling the tracking strand and excluding the non-tracking strand from the central channel. (**B**) CMG-forked DNA structure (PDB ID# 5U8S). Duplex DNA penetrates the zinc fingers defining the start of the central channel in the N-tier and the unwinding point is internal to the central channel (**Georgescu et al., 2017**). The Mcm2-7 subunits are shown in two shades of blue to highlight the N- and C-tiers. The lagging strand tail is present but not visible, indicating mobility, and is suggested to exit the central channel though a gap between two zinc fingers (dashed line) (see also panel C). (**C**) Illustration of CMG encountering a lagging strand block in the modified steric exclusion mode, partially encircling both strands of the duplex DNA as shown in (**B**). To bypass the block, CMG-DNA must isomerize to the classic steric exclusion mode in (**A**) or by opening of the Mcm2-7 ring during bypass (illustrated in Figure 8 of **Langston and O'Donnell, 2017**).

DOI: https://doi.org/10.7554/eLife.29118.002

The eukaryotic helicase is the Mcm2-7 heterohexamer which requires five additional accessory proteins - Cdc45 and the four subunit GINS complex – for full activity. This 11-subunit assembly is referred to as CMG (Cdc45, Mcm2-7, GINS) (**Ilves et al., 2010**; **Moyer et al., 2006**). The recent cryo-EM 3D structure of S. cerevisiae CMG helicase at a DNA replication fork shows a unique DNA binding feature (**Figure 1B**) (**Georgescu et al., 2017**). Instead of encircling only ssDNA, the zinc fingers of the N-tier encircle and bind the dsDNA and the DNA unwinding point is buried inside the central channel; the unwound leading strand then proceeds through the central channel into the C-tier motor domain. The lagging strand is not visualized in the structure, indicating mobility, and is proposed to enter the initial part of the central channel defined by the N-terminal zinc fingers and then exit the surface of CMG, through a gap between two zinc fingers. The dsDNA is held at a 28° angle to the central channel, surrounded by the zinc fingers at the 'top' of CMG. The dsDNA appears to be tightly held because if the CMG-dsDNA contact was flexible the DNA would have been averaged out during 3D reconstruction.

The structural evidence that S. cerevisiae CMG encircles dsDNA during unwinding is supported by recent biochemical experiments using strand specific dual streptavidin blocks that show CMG is halted by a block placed on either the non-tracking (lagging) or the tracking (leading) strand (**Figure 1C**) (**Langston and O'Donnell, 2017**). Interestingly, given a sufficiently long time S. cerevisiae CMG can proceed through the lagging strand block without displacing the streptavidin, suggesting that CMG slowly isomerizes to a classic steric exclusion mode that only encircles the tracking (leading) strand or isomerizes by ring opening to bypass the block (**Langston and O'Donnell, 2017**). Earlier studies in Xenopus extracts demonstrate that replisome progression is not hindered by a dual streptavidin block on the lagging strand and conclude that CMG functions in a steric exclusion mode by only encircling the leading strand (**Fu et al., 2011**). Considering that the zinc fingers of isolated S. cerevisiae CMG encircle dsDNA, we proposed that Xenopus extracts contain a factor that facilitates isomerization of CMG on DNA such that it can bypass blocks on the non-tracking strand (**Langston and O'Donnell, 2017**).

The present study identifies Mcm10 as the factor that enables CMG to rapidly bypass a block on the lagging strand. Mcm10 is unique to eukaryotes and is an essential gene product that, when mutated, causes abortive entry into S-phase (**Du et al., 2012**; **Merchant et al., 1997**;

*Solomon et al., 1992*; *Thu and Bielinsky, 2013*). Mcm10 has two internal DNA binding sites (*Du et al., 2012*; *Robertson et al., 2008*; *Warren et al., 2008*) and is known to bind Mcm2-7 and Cdc45 (*Christensen and Tye, 2003*; *Di Perna et al., 2013*; *Douglas and Diffley, 2016*; *Perez-Arnaiz et al., 2016*). Mcm10 is generally thought to act as an initiation factor (*Thu and Bielinsky, 2013*). The licencing of an origin in G1 phase involves several proteins to form a head-to-head double hexamer of Mcm2-7. Additional initiation factors transform the Mcm2-7 double hexamer into two head-to-head CMGs during S phase (*Bell and Labib, 2016*) but Mcm10 is not required for this action (*Kanke et al., 2012*; *van Deursen et al., 2012*; *Watase et al., 2012*; *Yeeles et al., 2015*). Mcm10 is only required at the last step of initiation and is proposed to either activate CMG helicase or help separate the two CMGs for bidirectional replication forks (*Kanke et al., 2012*; *Lõoke et al., 2017*; *Quan et al., 2015*; *van Deursen et al., 2012*; *Watase et al., 2012*; *Yeeles et al., 2015*). Additional studies indicate that Mcm10 may also function at replication forks (*Chadha et al., 2016*; *Gambus et al., 2006*; *Lõoke et al., 2017*; *Ricke and Bielinsky, 2004*).

The current study demonstrates that Mcm10 forms an isolable stoichiometric complex with CMG that greatly stimulates CMG helicase activity (up to 30-fold) and enhances processivity. Strikingly, Mcm10 uniquely promotes CMG bypass of a lagging strand block without displacing it, suggesting that Mcm10 isomerizes the CMG-DNA complex either to a steric exclusion mode or by facilitating ring opening. In contrast, leading strand blocks inhibit CMG unwinding but Mcm10 helps CMG displace the blocks and continue unwinding, consistent with the idea that Mcm10 makes CMG a more powerful, processive helicase. Despite the strong stimulation of helicase activity, Mcm10 has little effect on CMG-dependent DNA synthesis by the replisome in vitro, particularly in the presence of the Mrc1-Tof1-Csm3 complex. However, the replisome becomes stalled by lagging strand blocks in the absence of Mcm10, indicating that CMG uses an internal unwinding mechanism even in the context of rapidly moving replisomes. Stalled replisomes are released by addition of Mcm10, further supporting the idea that Mcm10 functions at the level of the helicase to enhance processivity, provide additional power and to enable blocked replisomes to bypass obstacles on the DNA. These functions may also explain the role of Mcm10 in origin initiation as described in the Discussion.

## Results

### Mcm10 forms a complex with CMG and greatly stimulates helicase activity

In *Figure 2A* we titrated Mcm10 into a helicase assay using a fixed amount of CMG and a synthetic forked DNA with a 50 bp duplex and 40-nt dT ssDNA tails. Stimulation by Mcm10 is remarkable, with over 30-fold enhancement of CMG unwinding at 2' in the presence of only one molecule of Mcm10 per CMG (compare lane 5 to lane 2 in *Figure 2A*). Reactions containing Mcm10 are complete by the 5 min time point, while the CMG reaction without Mcm10 continues slow unwinding during the 10 min time course (*Figure 2B*). The results at different Mcm10 concentrations indicate that maximal stimulation of CMG is observed with two subunits of Mcm10 for each CMG complex and further addition of Mcm10 beyond this amount has little effect. (*Figure 2C*). Control reactions with Mcm10 alone, lacking CMG, show no unwinding activity (*Figure 2—figure supplement 1*). Reactions with Mcm10 sometimes give a supershift of some of the substrate (e.g. lanes 5, 8, 11, 12 in *Figure 2A*) which we interpret as a gel shift of Mcm10 that remains bound to DNA, as Mcm10 is a known DNA binding protein (*Du et al., 2012*; *Robertson et al., 2008*; *Warren et al., 2008*). To determine if other DNA binding proteins such as RPA or E. coli SSB stimulate CMG, we performed experiments in which CMG unwinding was initiated and then either Mcm10, RPA or SSB was added. The results show that the stimulation of CMG is specific to Mcm10 and that RPA and SSB do not substitute for Mcm10 (*Figure 2—figure supplement 2*).

To determine if Mcm10 binds to CMG in a stable fashion, we mixed a 3-fold molar excess of Mcm10 (as monomer) with FLAG-tagged CMG and then isolated the CMG-Mcm10 complex using FLAG antibody magnetic beads (*Figure 2D*). The reaction was washed twice with buffer containing 300 mM NaCl and CMG-Mcm10 complex was eluted using FLAG peptide. The eluted material was analyzed by SDS/PAGE, and stoichiometric levels of Mcm10 were clearly visible with the Mcm2-7 subunits of CMG (lane 3). The GINS subunits of the CMG-Mcm10 complex ran off the gel, but a previous study demonstrated that Mcm10 preferentially associates with the entire CMG complex

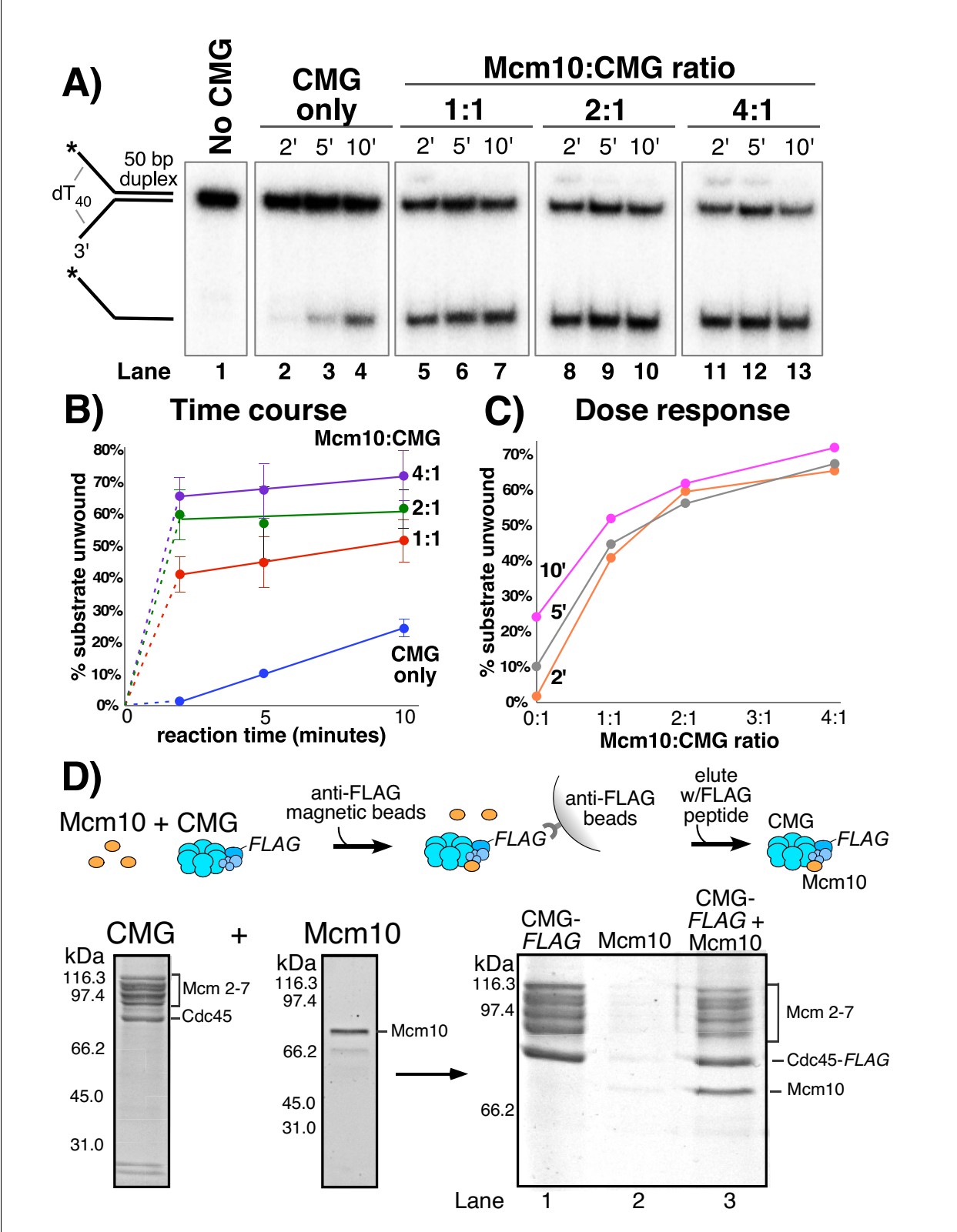

**Figure 2.** Mcm10 binds CMG and stimulates its helicase activity. (**A**) Titration of Mcm10 in a CMG unwinding assay. Reactions contained 25 nM CMG with either: no Mcm10 (lanes 2–4), 25 nM Mcm10 (1:1, lanes 5–7), 50 nM Mcm10 (2:1, lanes 8–10) or 100 nM Mcm10 (4:1, lanes 11–13). See Materials and methods for details. Also see *Figure 2—figure supplements 1–2*. (**B**) Quantification of the data from (**A**). Values are the average of three independent experiments and the error bars show the standard deviation. (**C**) The same data displayed as a dose-response curve of unwinding as a function of the

*Figure 2 continued on next page*

*Figure 2 continued*

Mcm10:CMG ratio in the reaction at the three different time points. (D) Isolation of CMG-Mcm10 complex using CMG-Flag and anti-Flag beads. CMG and a 3-fold molar excess of Mcm10 were mixed, attached to beads, then washed twice and eluted with Flag peptide. The PAGE gel shows elution from the anti-Flag beads that were loaded with either CMG-flag alone (lane 1), Mcm10 alone (lane 2), or the CMG-flag +Mcm10 mixture (lane 3). The protein bands are identified to the right. A densitometric scan of the bands in the gel of the CMG-Mcm10 complex indicates a molar ratio of 1.8 Mcm10 per Mcm2-7 complex; the Cdc45 is typically in excess due to the Flag tag (i.e. any dissociated Cdc45 will bind the beads). The GINS subunits ran off the gel (see *Figure 2—figure supplements 3–4* for GINS stoichiometry). Also see *Figure 2—figure supplement 5* for helicase activity of purified CMG-Mcm10 complex.

DOI: https://doi.org/10.7554/eLife.29118.003

The following figure supplements are available for figure 2:

**Figure supplement 1.** Mcm10 does not have helicase activity.

DOI: https://doi.org/10.7554/eLife.29118.004

**Figure supplement 2.** Neither yeast RPA nor E. coli SSB stimulates CMG helicase.

DOI: https://doi.org/10.7554/eLife.29118.005

**Figure supplement 3.** SDS PAGE of flag bead eluted CMG-Mcm10.

DOI: https://doi.org/10.7554/eLife.29118.006

**Figure supplement 4.** MonoQ reconstitution of CMG/Mcm10 complex.

DOI: https://doi.org/10.7554/eLife.29118.007

**Figure supplement 5.** The CMG-Mcm10 complex reconstituted on MonoQ is functional.

DOI: https://doi.org/10.7554/eLife.29118.008

(*Lõoke et al., 2017*). This is supported by densitometry analysis of gels showing GINS subunits in both FLAG and MonoQ purified CMG-Mcm10 complexes (*Figure 2—figure supplements 3* and *4*). Mcm10 was not visible in a control reaction in which Mcm10 was present with beads in the absence of CMG (*Figure 2D*, lane 2). We also found that the CMG-Mcm10 complex could be isolated using a MonoQ column and elutes at ~400 mM NaCl (*Figure 2—figure supplement 4*); densitometric scans are consistent with 1–2 Mcm10 per 1 of each CMG subunit (see legend to *Figure 2—figure supplement 4* for details). The reconstituted complex is functional in unwinding assays and shows much greater activity on the forked substrate than CMG alone (*Figure 2—figure supplement 5*), comparable to that seen when adding Mcm10 directly to the unwinding assay in *Figure 2*. The fact that Mcm10 associates with CMG sufficiently tightly to be isolated by MonoQ chromatography that elutes at 400 mM NaCl and by immunoaffinity beads using 300 mM salt washes suggests that CMG-Mcm10 is a stable complex. Densitometric scans from several preparations of CMG-Mcm10 made using both methods give a stoichiometry of ~1–2 Mcm10 subunits for each CMG complex, consistent with the Mcm10 titration results of *Figure 2A*.

## Mcm10 enhances the processivity of CMG helicase

The stimulation of unwinding by Mcm10 observed in *Figure 2* could be attributable to more efficient loading of CMG onto the substrate, faster unwinding, and/or greater processivity of unwinding. To distinguish among these possibilities, we compared CMG unwinding of a fork with a longer, 160 bp duplex region to that of the fork with a 50 bp duplex (as in *Figure 2*) in the presence and absence of Mcm10 (*Figure 3*). The two substrates have identical 3' and 5' ssDNA tails, so loading of CMG onto the forks should be the same and any differences in unwinding should be attributable to differences in CMG unwinding activity over the different lengths of duplex. For these experiments, CMG was pre-incubated with the DNA substrate for 10' followed by addition of ATP ± Mcm10 to start the reaction (reaction scheme in *Figure 3A*). In the absence of Mcm10, substantial differences were observed in unwinding of the two substrates that suggest limited processivity of the helicase, consistent with reports of low processivity by Drosophila and human CMG (*Kang et al., 2012*; *Moyer et al., 2006*). CMG unwound only 3% of the longer 160 bp duplex fork in 10' (*Figure 3B* lanes 1–10 and graph in *Figure 3D*) compared to 24% for the shorter 50 bp duplex fork at 10' (*Figure 3C*). Even after 30' only 8% of the 160 bp duplex was unwound by CMG indicating that the difference in activity between the two substrates is attributable to low processivity of CMG rather than simply the additional time it takes to unwind the longer substrate.

The experiments were repeated in the presence of Mcm10 at a 2:1 ratio to CMG (as determined in *Figure 2A–C*). Surprisingly, in the presence of Mcm10 the unwinding curves of the two substrates

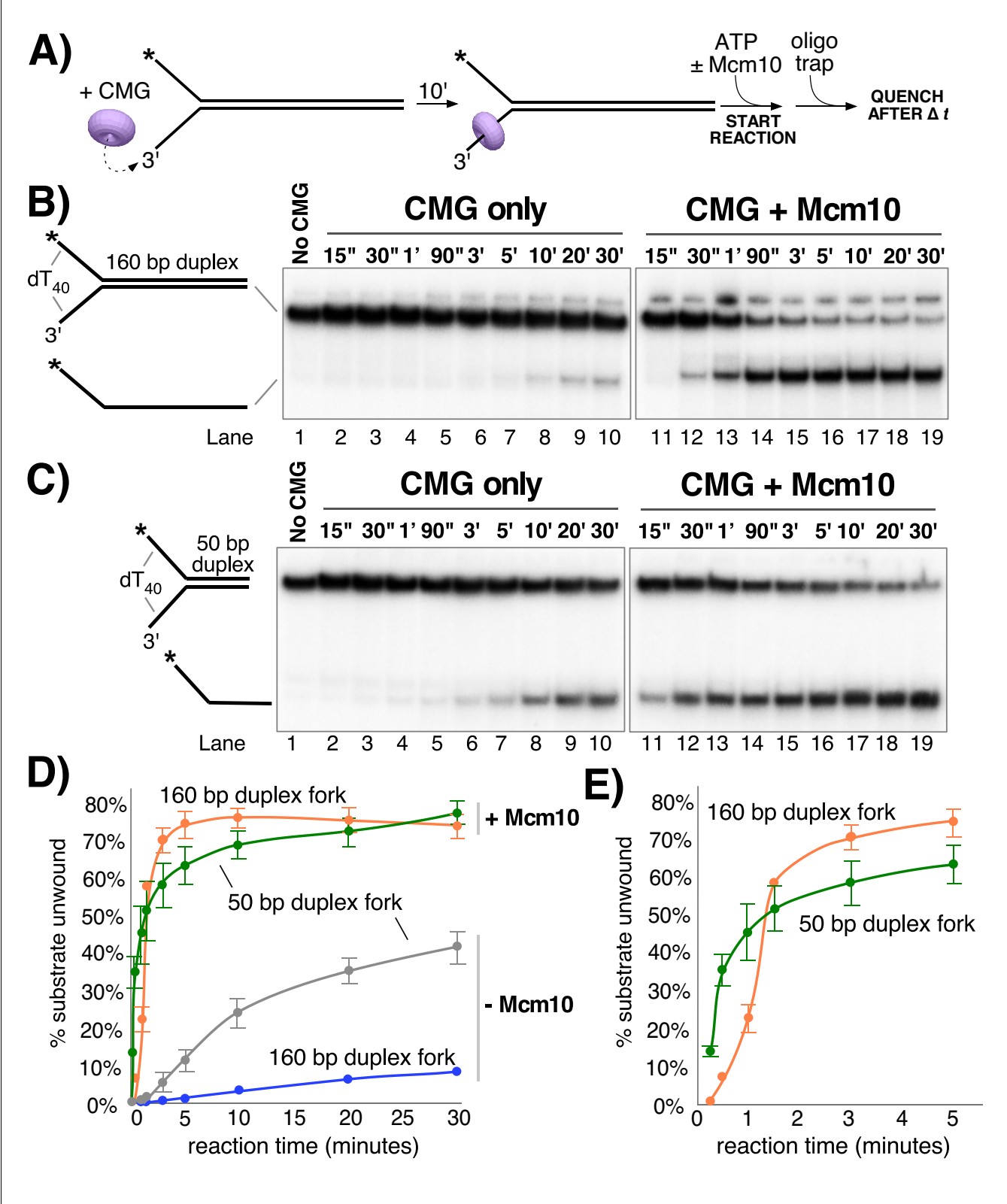

**Figure 3.** Mcm10 enhances the processivity of CMG unwinding. (A) Scheme of the reaction. CMG was pre-incubated with the substrates for 10′ before addition of ATP ± a 2-fold excess of Mcm10 over CMG. (B) Native PAGE analysis of CMG unwinding in the absence (lanes 2–10) or presence (lanes 11–19) of Mcm10 using either a 160 bp duplex fork (panel B) or a 50 bp duplex fork (panel C) as indicated by the schematics to the left of the gels. (D)

*Figure 3 continued on next page*

*Figure 3 continued*

Quantification of the data in the gels. (**E**) The first 5′ of reactions with Mcm10 are shown to better illustrate the effect of Mcm10 on unwinding by CMG. The values are averages of three independent experiments and the error bars show the standard deviation. Also see *Figure 3—figure supplement 1*.

DOI: https://doi.org/10.7554/eLife.29118.009

The following figure supplement is available for figure 3:

**Figure supplement 1.** Delayed replication of longer duplex allows estimation of unwinding rate.

DOI: https://doi.org/10.7554/eLife.29118.010

were very similar, in contrast to experiments in the absence of Mcm10 (*Figure 3B and C*, lanes 11–19 and graph in *Figure 3D*). This result suggests that Mcm10 enhances the processivity of CMG and possibly also stimulates the rate of CMG unwinding. Examination of the first 5 min of unwinding in the presence of Mcm10 shows a short delay in the appearance of products using the 160 bp duplex compared to the 50 bp duplex (*Figure 3E*). We took advantage of this delayed appearance of the longer products to design an experiment that approximates the average rate of unwinding by CMG-Mcm10 (*Figure 3—figure supplement 1*). CMG was pre-incubated with an equimolar mixture of the two different length substrates and then the reaction was started by the addition of ATP and Mcm10. The substrates and products migrate at distinguishable positions in a native PAGE gel allowing us to observe the appearance of both products in the same reaction. As shown in *Figure 3—figure supplement 1*, appearance of unwound 160 bp duplex products was delayed by ~46 s compared to the 50 bp duplex, indicating an average rate of unwinding of 2.4 bp/s over the additional 110 bp. The final extent of unwinding is similar for both substrates, so the delay reflects a difference in time of unwinding the two DNAs due to their difference in length rather than a difference in helicase activity on the two substrates.

## Mcm10 stimulates CMG-dependent leading strand replication in the absence of Mrc1-Tof1-Csm3

Having observed stimulation of CMG by Mcm10 in helicase assays, we wished to determine whether Mcm10 also stimulated CMG in the context of the DNA replication fork. To do so, we used a 2.8 kb linear forked DNA primed with a $^{32}$P-5′ end-labeled 37mer oligo (*Georgescu et al., 2014*; *Langston et al., 2014*) and tested the effect of Mcm10 on extension of the radiolabeled primer by the core leading strand replisome consisting of CMG, Pol ε, RFC, PCNA, and RPA (*Figure 4*). CMG is pre-incubated with the DNA substrate (in the presence or absence of Mcm10) for 5′ at 30° C to allow CMG loading onto the 3′ tail of the fork (see reaction scheme in *Figure 4A*). ATP is omitted at this stage to prevent CMG from unwinding the DNA before assembly of the replisome. Pol ε, RFC, PCNA and 2 dNTPs are then added and incubated a further 4′ to assemble the core leading strand replisome and the reaction is started upon addition of the remaining dNTPs and ATP along with RPA. Analysis of the autoradiogram in *Figure 4B* shows a modest enhancement of CMG-dependent primer extension by Mcm10. In the presence of Mcm10 the 2.8 kb full length product can be observed at the 6′ time point (*Figure 4B*, lanes 8 and 13), yielding a maximum fork rate of 7.8 nucleotides (ntds)/s; 467 ntds/min), while 8′ is required to observe the full-length product in the absence of Mcm10 (*Figure 4B*, lane 4; 5.8 ntds/s; 350 ntds/min). This result is consistent with modest stimulation of the rate of DNA synthesis by Mcm10 in an origin-dependent plasmid replisome assembly assay (*Lõoke et al., 2017*).

Cell biological and genetic studies have shown that replication forks move about 2-fold slower in cells that lack Mrc1 (*Hodgson et al., 2007*; *Petermann et al., 2008*; *Szyjka et al., 2005*; *Tourrière et al., 2005*) and these observations have been recapitulated in vitro in a origin plasmid replication system (*Yeeles et al., 2017*). The in vitro results demonstrated that Mrc1 stimulation of fork speed was most efficient in the presence of a Tof1-Csm3 heterodimeric complex. The stimulation by Mrc1 ± Tof1-Csm3 in the origin plasmid based assay did not require Ctf4, FACT, TopI or lagging strand Pol δ extension (*Yeeles et al., 2017*). Mcm10 is essential for initiation in the plasmid assay system and this requirement presents difficulty in determining whether Mcm10 also enhanced the rate of replication fork progression when Mrc1-Tof1-Csm3 was present (*Yeeles et al., 2017*). In our experiments, CMG is added as a purified complex enabling a direct test of whether Mcm10 stimulates fork speed in the presence of these additional factors. Hence we expressed and purified a

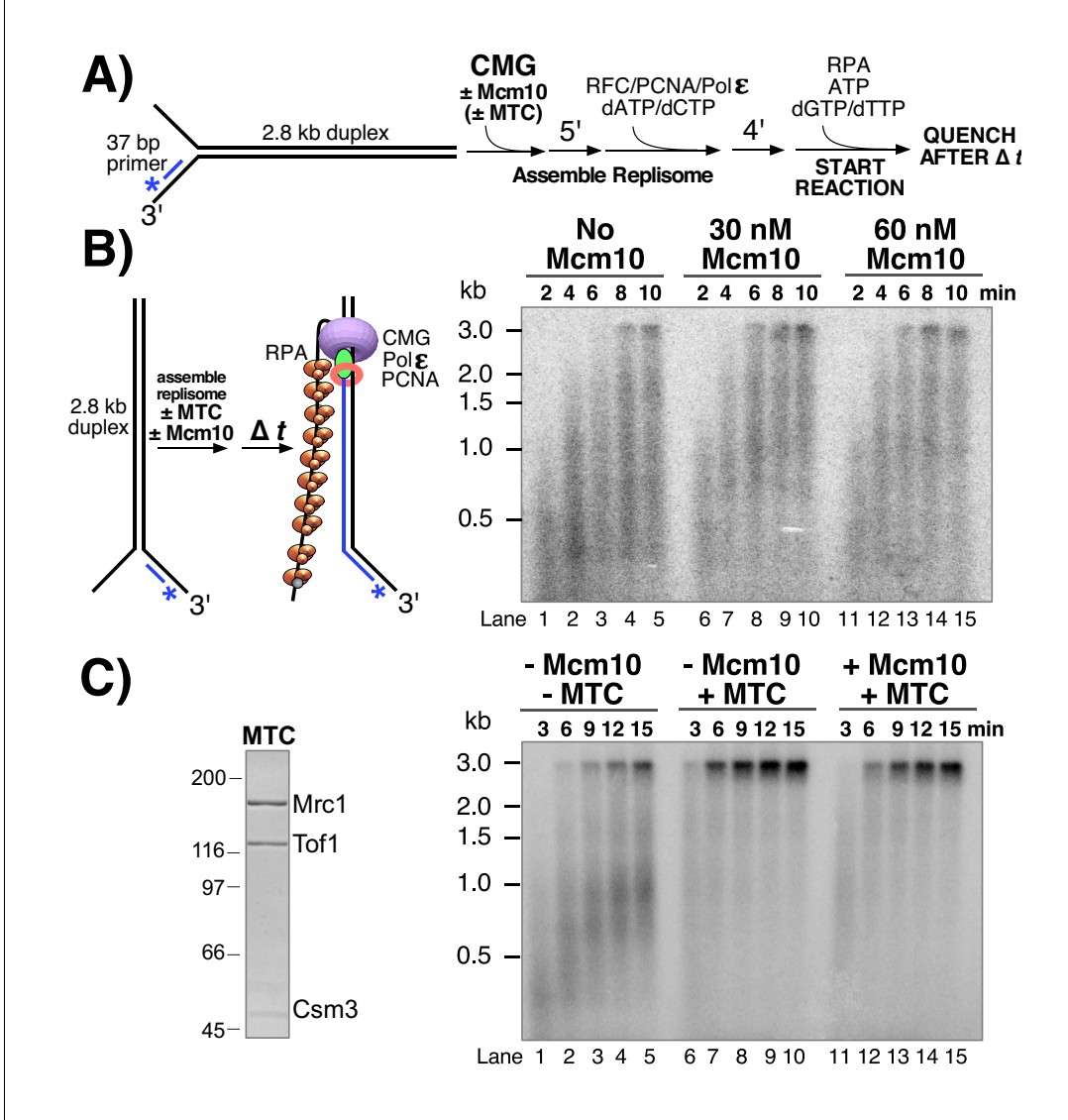

**Figure 4.** Mcm10 stimulates CMG-dependent leading strand replication in the absence of Mrc1-Tof1-Csm3. (A) Reaction scheme. See Materials and methods for details. (B) Alkaline agarose gel of leading strand products synthesized by the core leading strand replisome with the indicated amounts of Mcm10. (C) Purified MTC complex is shown to the left. Numbers to the left of the SDS PAGE gel show the migration of molecular weight markers (kDa). Time courses, to the right, are the core leading strand replisome with or without Mcm10 and Mrc1-Tof1-Csm3 complex (MTC) as indicated above the gel.

DOI: https://doi.org/10.7554/eLife.29118.011

Mrc1-Tof1-Csm3 heterotrimer, referred to herein as MTC complex, and tested its effect on the leading strand replisome with and without Mcm10 (*Figure 4C*). The autoradiogram shows that in the presence of the MTC complex, full length 2.8 kb product can be observed after 3 min (*Figure 4C*, lane 6), indicating a rate of 20 ntds/s (933 ntds/min), close to the 1.5–1.9 kb/min rate of replisomes in vivo (*Conti et al., 2007*; *Hodgson et al., 2007*; *Sekedat et al., 2010*). Furthermore, the MTC complex clears up the many immature products formed in the absence of MTC, suggesting that replication forks that stop or pause are brought to full length by the MTC complex. Addition of Mcm10 to leading strand replisome reactions containing the MTC complex does not result in a faster replication fork rate under the conditions used and may have even slightly slowed the replisome (*Figure 4C*, lanes 11–15). Thus, the improved rate effect of Mcm10 on the core leading strand replisome observed in *Figure 4B* is eclipsed by the rate enhancement provided by the MTC complex.

## Mcm10 promotes CMG unwinding past a lagging strand block

Together, the results of *Figures 2* and *3* show that Mcm10 strongly stimulates CMG unwinding, yet the experiments of *Figure 4* indicate that Mcm10 is not essential for normal replisome progression in vitro, particularly in the presence of the MTC complex. To explain these apparently conflicting results, we hypothesized that Mcm10 is required for replisome progression in particular circumstances, for example when CMG is hindered by DNA-bound proteins or other structural impediments on DNA. Indeed, Mcm10 is required for efficient initiation of DNA unwinding at an origin, a situation in which the two CMG complexes block one another's progression because they must pass each other in order to establish two bidirectional forks (see Discussion) (*Georgescu et al., 2017*).

To test the hypothesis that Mcm10 may help CMG overcome blocks, we examined the effect of Mcm10 on CMG unwinding of forked DNA with strand-specific blocks on either the lagging or leading strand of the duplex portion of the DNA (*Figure 5* and *Figure 5—figure supplement 1*, respectively). In these experiments, the block is formed by two biotinylated nucleotides on one DNA strand or the other, spaced 6 bp apart, that are tightly bound by the 53 kDa streptavidin protein. We recently showed that CMG unwinding is inhibited by biotin-streptavidin on either strand and is nearly inactive when two biotin-streptavidin blocks are present on either strand of the duplex (*Langston and O'Donnell, 2017*). This is unique to CMG, as thus far all other hexameric helicases examined are not blocked by an obstruction on the non-tracking strand and are only blocked by an obstruction on the translocating strand (*Hacker and Johnson, 1997*; *Kaplan, 2000*; *Kaplan et al., 2003*; *Lee et al., 2014*; *Nakano et al., 2013*). Inhibition of hexameric helicases by a block on one strand but not the other is interpreted as a steric exclusion mode of unwinding in which the non-tracking strand is completely excluded from the central channel of the helicase ring, thereby explaining why bulky substituents on the non-tracking strand do not affect helicase unwinding.

In contrast, S. cerevisiae CMG is inhibited by a block on either the non-tracking strand or the tracking strand (*Langston and O'Donnell, 2017*), consistent with the CMG-forked DNA structure showing that CMG encircles both strands of the forked junction (see *Figure 1B–C*) (*Georgescu et al., 2017*). However, when given sufficient time, the bulk of the CMGs that progress past the lagging strand block leave streptavidin attached to DNA and thus slowly isomerize on DNA to bypass the block without displacing it (*Langston and O'Donnell, 2017*). Isomerization for bypass could be achieved by CMG encircling only the leading strand or the Mcm2-7 ring within CMG may crack open for bypass, as detailed later in the Discussion.

In contrast to studies with pure CMG, studies of replication in Xenopus egg extracts show that two adjacent streptavidin blocks spaced 5 bp apart on the non-translocating (lagging) strand are quickly bypassed (within 10 min) by the replisome (*Fu et al., 2011*). The fact that our earlier study showed that isolated CMG is strongly inhibited by a dual streptavidin block, suggests that some other factor in the complete extract helps CMG bypass lagging strand blocks (*Langston et al., 2017*). To test whether Mcm10 may be the factor that facilitates CMG bypass of a lagging strand block we examined CMG unwinding of the forked DNA substrate with two biotinylated nucleotides on the duplex portion of the lagging strand template (see *Figure 5A*). Control reactions lacking Mcm10 show that CMG helicase is essentially shut down by the streptavidin blocks (*Figure 5B*, compare lanes 2–4 without streptavidin to lanes 5–7 with streptavidin) as previously observed (*Langston and O'Donnell, 2017*). But surprisingly, when Mcm10 is added, the rate and amount of unwinding are the same in the presence and absence of streptavidin (*Figure 5B*, compare lanes 8–10 and 11–13). Hence, Mcm10 enables CMG to rapidly bypass or displace a dual streptavidin block on the lagging strand. This observation mirrors results observed in the Xenopus system, which was assumed (but not demonstrated) to bypass these blocks without displacement. Ability of CMG to bypass lagging strand blocks with Mcm10 present in the pure protein system resolves the discrepancies between use of purified CMG (without Mcm10), and observations in Xenopus extracts (*Fu et al., 2011*; *Langston and O'Donnell, 2017*). We revisit the issue of whether streptavidin blocks are bypassed or displaced later in this report.

Having previously shown that the MTC complex greatly stimulates progression of the leading strand replisome even in the absence of Mcm10 (*Figure 4*), we tested the MTC complex in CMG helicase assays to see if it enhances the ability of CMG to bypass blocks on DNA (*Figure 5C*). In contrast to Mcm10, MTC was unable to promote CMG bypass of the block (compare lanes 11–13 in *Figure 5C* to lanes 11–13 in *Figure 5B*). We also note that MTC did not stimulate CMG unwinding

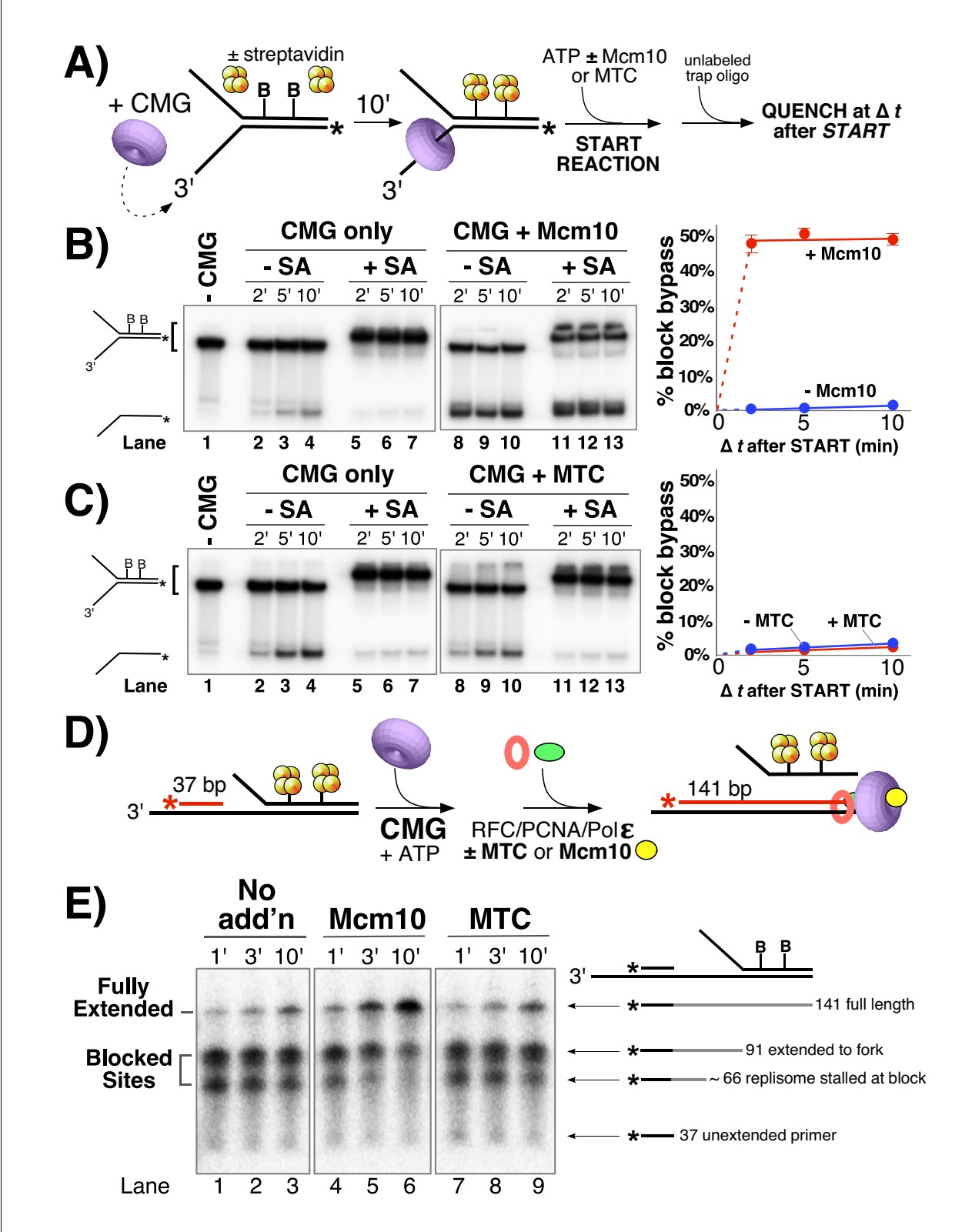

**Figure 5.** Mcm10, but not MTC complex, enables CMG and the replisome to bypass a lagging strand block. (**A**) Illustration of the helicase reaction scheme. See Materials and methods for details. (**B**) Unwinding assays on dual biotinylated forks with or without streptavidin and with or without Mcm10, as indicated. (**C**) Same as in panel B except reactions contained or omitted MTC complex instead of Mcm10. (**D**) Illustration of the replisome reaction scheme. Reactions contained a synthetic minifork with $^{32}$P-5' end-labeled primer and a dual biotin-streptavidin (SA) block on the lagging strand

*Figure 5 continued on next page*

*Figure 5 continued*

template. CMG and ATP were added for 3 min to allow CMG to bind DNA and translocate to the SA blocks, then Pol ε, RFC and PCNA were added (±MTC) along with all four dNTPs to assemble and extend the primer up to the blocked CMG, followed by addition of Mcm10 (or no protein). See Materials and methods for details. (E) Timed aliquots after ± Mcm10 addition were analyzed in a 10% urea PAGE gel. The primer, two stalled products and the full-length product are indicated to the right of the gel. Also see *Figure 5—figure supplement 1*.

DOI: https://doi.org/10.7554/eLife.29118.012

The following figure supplement is available for figure 5:

**Figure supplement 1.** A leading strand block inhibits CMG unwinding even in the presence of Mcm10.

DOI: https://doi.org/10.7554/eLife.29118.013

even in the absence of the block (*Figure 5C* compare lanes 8–10 with MTC to lanes 2–4 without MTC), suggesting that MTC does not function at the level of CMG unwinding. Together with the results of *Figures 2–4*, these data suggest that Mcm10 and MTC affect fork progression in very different ways. In contrast to Mcm10 mediated bypass of lagging strand blocks, CMG unwinding was delayed and strongly repressed by a dual streptavidin block on the leading strand even with Mcm10 present (*Figure 5—figure supplement 1*).

## Mcm10 is needed at the replication fork to bypass lagging strand barriers

To further investigate the functions of Mcm10 and MTC at a replication fork, we adapted the helicase substrate from *Figure 5A–C* to support DNA replication. The 3' tail of the fork was extended to accommodate a primer for DNA synthesis and the reaction was staged as illustrated in *Figure 5D*. First, CMG was added with ATP to allow helicase translocation to the block, followed by Pol ε/RFC/PCNA/dNTPs to see if the 'push' on CMG from a Pol ε-PCNA motor can overcome the block. The results in *Figure 5E* lanes 1–3 show that Pol ε-PCNA extended the $^{32}$P-primer to the stalled CMG (~66 bp) but had limited ability to bypass the block to form the 141 bp full-length product. The ~91 bp bands result from forks lacking CMG, so Pol ε-PCNA extends to the fork but cannot proceed further because Pol ε lacks strand displacement capability and relies on CMG to unwind the DNA for leading strand synthesis (*Ganai et al., 2016*; *Georgescu et al., 2014*). MTC has little or no effect on block bypass by the replisome (*Figure 5E* lanes 7–9), and is comparable to Pol ε-PCNA-CMG in lanes 1–3. By contrast, addition of Mcm10 allows the replisome to efficiently and rapidly proceed past these blocks as evidenced by the appearance of full length 141 bp product in lanes 4–6. These results confirm that Mcm10 is needed for replisome bypass of a block and they also indicate that neither MTC nor Pol ε-PCNA is able to drive the forward movement of a stalled CMG in the absence of Mcm10.

## Lagging strand blocks are bypassed while leading strand blocks are displaced

We next addressed whether Mcm10 enables CMG to bypass the lagging strand streptavidin or whether the apparent stronger helicase activity of CMG helicase provided by Mcm10 enables CMG to displace the lagging strand blocks (*Figure 6*). To determine if the streptavidin was displaced or retained on DNA during CMG unwinding we $^{32}$P-labeled the biotinylated strand, which is gel shifted by streptavidin, and repeated the assay in the presence of excess biotin which traps displaced streptavidin and prevents it from rebinding biotinylated DNA. The control reactions, adding the biotin trap before streptavidin, show >96% inhibition of the streptavidin gel shift of the biotinylated DNAs (*Figure 6—figure supplement 1*). Upon performing the experiment using dual biotinylated-streptavidin blocks on the lagging strand template, the results show that nearly all of the unwound product DNA retains the streptavidin (i.e. it is gel shifted) in the presence of the excess biotin trap. Hence, CMG-Mcm10 bypasses lagging strand streptavidin blocks without displacing them from DNA.

We also examined the effect of Mcm10 on CMG encounter with dual biotin-streptavidin blocks on the leading strand. We showed earlier (*Figure 5—figure supplement 1*) that CMG is unable to unwind the DNA in the presence of streptavidin and that while CMG-Mcm10 is also strongly inhibited, it unwinds a detectable amount of the DNA within 10 min. To test whether the observed unwinding by CMG-Mcm10 through leading strand blocks is due to streptavidin bypass or

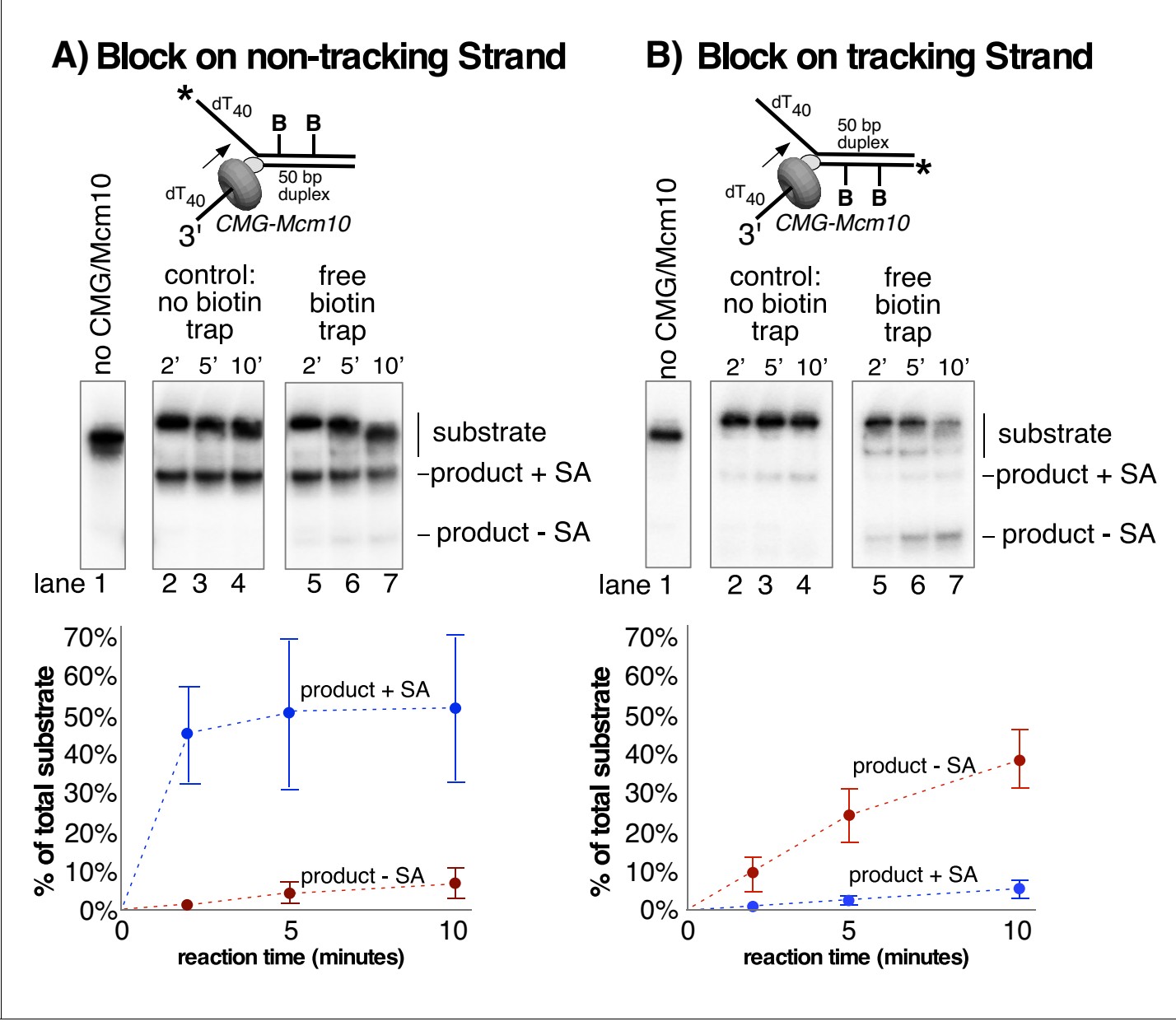

**Figure 6.** Lagging strand blocks are not displaced from DNA during unwinding by CMG-Mcm10. (**A**) The helicase reaction in *Figure 5B* (CMG + Mcm10) was repeated with the radiolabel on the biotinylated lagging strand and in the presence of free biotin as a trap for displaced streptavidin (See Materials and methods for details). Unwound strands are shifted in the gel when bound to streptavidin (product +SA) compared to unbound (product – SA). Lanes 2–4 are a control in the absence of biotin trap showing that all of the unwound products migrate in the gel shifted position. The biotin trap is present in the experiment in lanes 5–7. A shift in migration of the unwound products indicates displacement of streptavidin by CMG-Mcm10, but the result shows that almost all the unwound DNA retained the streptavidin. (**B**) The experiment in (**A**) was repeated with a substrate in which the dual biotin blocks were on the radiolabeled leading strand. Also see *Figure 6—figure supplement 1*.

DOI: https://doi.org/10.7554/eLife.29118.014

The following figure supplement is available for figure 6:

**Figure supplement 1.** Free biotin prevents streptavidin binding to the biotinylated oligos but does not displace pre-bound streptavidin.

DOI: https://doi.org/10.7554/eLife.29118.015

streptavidin displacement we repeated the experiment with the [32]P-label on the biotinylated strand and using the excess biotin trap. The results demonstrate that streptavidin is displaced from the unwound product (*Figure 6B*). Hence, Mcm10 appears to provide CMG with more translocation force than CMG alone, consistent with the increase in processivity of CMG in the presence of Mcm10 (*Figure 3*).

## Discussion

We have shown that a stoichiometric CMG-Mcm10 complex can be reconstituted and isolated (*Figure 2D* and *Figure 2—figure supplements 3–5*) and that Mcm10 greatly stimulates CMG unwinding and processivity (*Figures 2* and *3*). Interestingly, the effect of Mcm10 on fork progression per se is relatively insignificant in the absence of blocks (*Figure 4*). Notably, we find that Mcm10 enables CMG and the replisome to bypass lagging strand blocks that otherwise bring unwinding and leading strand synthesis to a halt (*Figures 5* and *6*). The function of Mcm10 in facilitating replisome bypass of lagging strand blocks is consistent with genetic studies showing that Mcm10 deficiency leads to genome instability and increased dependence on post-replication and recombinational repair pathways to maintain viability (*Araki et al., 2003*; *Lee et al., 2010*).

Previous reports have solved the structure of the internal DNA binding region of Xenopus Mcm10 and documented that the internal region has two DNA binding elements, an OB fold and a zinc finger, that allow it to bind ssDNA and/or dsDNA (*Du et al., 2012*; *Robertson et al., 2008*; *Warren et al., 2008*). Following the internal DNA binding region is a C-terminal region that forms the main interaction site to Mcms (*Douglas and Diffley, 2016*; *Lõoke et al., 2017*; *Quan et al., 2015*). Furthermore, Mcm10 appears to bind to the N-tier region of the CMG complex that faces the forked junction (*Figure 1B*), as recent studies show that interaction of Mcm10 with Mcm2-7 is mediated primarily through an interaction with a conserved domain in the N-terminal region of the Mcm2 subunit (*Lõoke et al., 2017*). Consistent with these known properties of Mcm10, we propose in *Figure 7* that, upon encountering a lagging strand block, Mcm10 binds at the N-tier of CMG and also binds DNA at the fork. This places Mcm10 between CMG and the forked junction such that duplex DNA no longer enters the zinc fingers that line the top of the central channel of CMG. In this

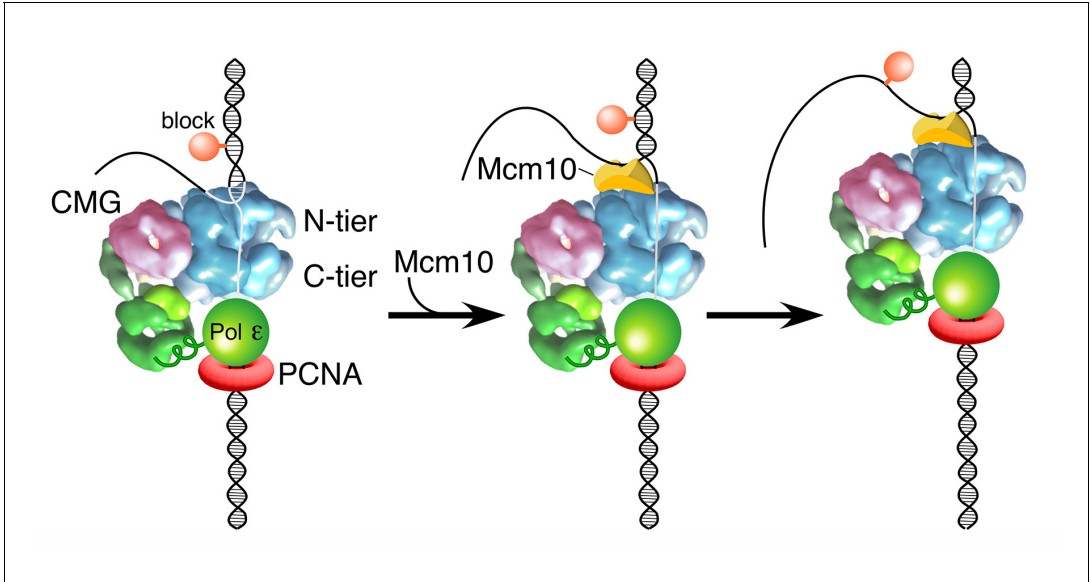

**Figure 7.** Model of Mcm10 function. Proposed model of Mcm10 mediated isomerization of CMG-DNA at a replication fork. Left, CMG encounters an impediment on the DNA but cannot pass it because it surrounds dsDNA in the N-terminal tier of Mcm2-7 (as in *Figure 1C*); middle, Mcm10 binds to the forked junction and to the N-terminal tier of CMG, releasing CMG's grip on the duplex DNA and placing CMG in a steric exclusion mode encircling ssDNA (as in *Figure 1A*); right, with the lagging strand freed from the interior of the Mcm2-7 ring, CMG can bypass the impediment on the DNA and proceed. Alternative isomerization reactions that could also accomplish bypass of a bulky block are discussed in the text.
DOI: https://doi.org/10.7554/eLife.29118.016

configuration, Mcm10 isomerizes CMG to a steric exclusion mode where it is capable of bypassing blocks on the non-tracking strand as observed with other hexameric helicases and with the complete replisome in the Xenopus extract system (*Fu et al., 2011*). Two alternative isomerization reactions that would also facilitate lagging strand block bypass without displacement include one in which Mcm10 cracks open the N-tier of the Mcm2-7 ring of CMG, bypasses the block, and recloses the N-tier. Similarly, Mcm10 might enable opening of both the N- and C-tiers, either simultaneously or in succession, to bypass the blocks. These alternative isomerization reactions are described in detail in (Figure 8 of *Langston et al., 2017*).

### Possible role of Mcm10 at the origin

Ability to rapidly bypass blocks on the non-tracking strand is inherent in all replicative hexameric helicases examined with the exception of eukaryotic CMG. One possible reason that CMG evolved a requirement for another protein to bypass a lagging strand block is for regulation of fork initiation and progression. It is documented from several laboratories that the head-to-head (N-to-N) double hexamer of Mcm2-7 at an origin matures to form two CMGs on dsDNA without unwinding DNA (*Heller et al., 2011*; *Kanke et al., 2012*; *van Deursen et al., 2012*; *Watase et al., 2012*; *Yeeles et al., 2015*). Mcm10 is required at the last step of origin initiation to 'activate' CMG to unwind DNA as detected by RPA binding to the ssDNA products. These observations can be explained by the findings of this report. Specifically, CMG is known to track with the N-tier ahead of the C-tier, and therefore head-to-head CMGs at an origin are directed inward toward one another. Hence, each CMG blocks the other CMG at an origin. However, if Mcm10 activates the CMGs to locally unwind DNA, as observed by head-to-head SV40 T-antigen hexamers that produce ssDNA 'rabbit ears' at the viral origin (*Wessel et al., 1992*), the two CMGs could isomerize to encircle opposite single-strands and pass one another to form bidirectional forks (see Figure 8B in [*Georgescu et al., 2017*]). This same action may also promote termination, when two forks meet, enabling the CMG based replisomes to easily pass one another.

### Is Mcm10 a stable component of a moving replisome?

Considering that Mcm10 does not appear to increase the elongation rate of the replisome (*Figure 4C*), it may not be required at all times. For example, Mcm10 might only be needed to help the replisome bypass blocks on the lagging strand which conceivably could be performed by a dynamic Mcm10 that interacts only transiently with a stalled replisome. Alternatively, Mcm10 mediated isomerization of CMG-DNA may remain stable, in which case Mcm10 may not be needed for some time after its first use, and in fact an Mcm10 effect on CMG that only requires transient Mcm10 binding has been demonstrated in an earlier study (*Lõoke et al., 2017*).

Mcm10 binds remarkably tightly to CMG, eluting as a CMG-Mcm10 complex from beads after high salt washes (*Figure 2D*), and the CMG-Mcm10 complex also remains associated and elutes from a MonoQ ion-exchange resin at high ionic strength (400–450 mM NaCl) (*Figure 2—figure supplement 4*). Additionally, some studies suggest Mcm10 is a component of moving replisomes (*Chadha et al., 2016*; *Gambus et al., 2006*; *Lõoke et al., 2017*; *Ricke and Bielinsky, 2004*). However, other cellular studies indicate that Mcm10 is not in constant association with the replisome (*Kanke et al., 2012*; *van Deursen et al., 2012*; *Watase et al., 2012*). Thus, while the salt stability of the CMG-Mcm10 complex reported here suggests it might remain attached to the replisome, its kinetics of attachment are not addressed here and Mcm10 could still be dynamic with CMG, or its on/off rates with the replisome could be affected by other proteins in the cell. Future structural and single-molecule studies of CMG-Mcm10 may be expected to shed light on further details about the dynamics of Mcm10 and the mechanism of its replisome bypass function at stalled replisomes described in this report.

## Materials and methods

### Reagents

Radioactive nucleotides were from Perkin Elmer and unlabeled nucleotides were from GE Healthcare. DNA modification enzymes were from New England Biolabs. DNA oligonucleotides were from Integrated DNA Technologies. Streptavidin was from Pierce/Thermo Scientific. 5 mg of streptavidin

powder was resuspended in 0.5 ml distilled water to make a 10 mg/ml stock. D-Biotin (50 mM aqueous solution) was from Invitrogen/Molecular Probes. Protein concentrations were determined using the Bio-Rad Bradford Protein stain using BSA as a standard. E. coli SSB and S. cerevisiae CMG, Pol ε, RFC, PCNA and RPA were overexpressed and purified as previously described (*Georgescu et al., 2014*; *Langston et al., 2014*).

MBP-Mcm10. Mcm10, with a Maltose Binding Protein (MBP) tag at the N-terminus, was purified by applying clarified extract from 12L *E. coli* overexpression cells at 1 ml/min to a 12 ml amylose column (New England Biolabs) in MBP buffer (20 mM Tris-Cl pH 7.5, 0.5 M NaCl, 1 mM DTT and 1 mM EDTA). The column was washed with 100 ml MBP buffer and eluted with 60 ml MBP buffer supplemented with 10 mM D-(+)-maltose (Sigma Aldrich) collecting 1.5 ml fractions. Peak fractions containing MBP-Mcm10 were pooled (~20 ml/85 mg) and dialyzed 2 hr vs. 2 L Buffer S (25 mM Hepes-KOH pH 7.5, 10% glycerol, 2 mM DTT, 2 mM MgCl$_2$, and 1 mM EDTA). The dialysate was spun at 5,000 rpm in a swinging bucket rotor in an RC3B centrifuge to remove any precipitated material. The supernatant was applied to a 10 ml SP-Sepharose column that had been previously equilibrated in Buffer S supplemented with 100 mM NaCl and washed with 50 ml of the same buffer. The column was eluted with a 100 ml linear gradient from 100 to 600 mM NaCl in Buffer S. Peak fractions containing full-length MBP-Mcm10 were dialyzed overnight vs. 2L buffer S/200 mM NaCl and re-applied to a second 10 ml SP-Sepharose column and eluted with a step gradient from 20% to 50% of the buffer in pump B in 5% increments (pump A contained buffer S/100 mM NaCl and pump B contained buffer S/600 mM NaCl). The peak of full-length MBP-Mcm10 eluted at 50% pump B and was pooled and dialyzed against buffer S/200 mM NaCl in the presence of PreScission Protease to remove the MBP tag. Dialyzed pools were aliquoted, flash frozen in liquid nitrogen and stored at −80°C.

His-FLAG Mcm10. Mcm10 with a hexahistidine tag at the N-terminus and a 3X FLAG tag at the C-terminus, was purified by first applying clarified extract from 72L *E. coli* overexpression cells to a column containing 10 ml Chelating Sepharose Fast Flow (GE Healthcare) charged with 50 mM NiSO$_4$. The column had been previously washed with binding buffer (20 mM Tris-Cl pH 7.9, 5 mM imidazole, 500 mM NaCl, 0.01% NP-40) and the *E. coli* extract was adjusted to approximate the ionic strength of the binding buffer and applied to the column at 1 ml/min. The column was washed with binding buffer and then eluted with the same buffer containing 375 mM imidazole. The eluted material was then applied to a column containing 6 ml ANTI-FLAG M2 Affinity Gel (Sigma) at 0.1 ml/min. The FLAG column had been previously equilibrated in FLAG buffer (20 mM Tris-Cl pH 7.5, 10% glycerol, 500 mM NaCl, 1 mM DTT, 1 mM MgCl$_2$, 0.01% NP-40) and after loading, the column was washed with 50 ml of FLAG buffer. The bound material was eluted with 20 ml FLAG buffer containing 0.2 mg/ml 3X FLAG peptide (EZ Biolab, Carmel, Indiana USA) and collecting 1.5 ml fractions. The elution buffer was applied to the column in two 6 ml increments pausing 30′ after each increment followed by 3 ml increments with 30′ pauses until the elution was complete. Eluted material was aliquoted, flash frozen in liquid nitrogen, and stored at −80°C.

Mrc1-Tof1-Csm3 complex. The 3 subunits of the MTC complex (Mrc1-Tof1-Csm3) were co-expressed in yeast. Genes were integrated into the chromosome of strain OY01 (ade2-1 ura3-1 his3-11,15 trp1-1 leu2-3,112 can1-100 bar1Δ MATa pep4::KANMX6) with Mrc1$^{Flag}$ integrated at the Ade2 locus, untagged Tof1 at the His3 locus, and $^{His}$Csm1 at the Leu2 locus, each under control of the Gal1/10 promoter. Cells were grown under selection at 30°C in SC glucose, then split into 18L YP-glycerol and grown to OD600 of 0.7 at 30°C before induction for 6 hr upon addition of 20 g of galactose/L. After 6 hr, cells were harvested by centrifugation, resuspended in a minimal volume of 20 mM HEPES, pH 7.6, 1.2% polyvinylpyrrolidone, and protease inhibitors and frozen by dripping into liquid nitrogen. Purification of MTC was performed by lysis of 18 L equivalent of frozen cells with a SPEX cryogenic grinding mill. Ground cell powder was thawed in the cold room and resuspended to 25 ml final volume with 5X FLAG binding buffer (1x is 250 mM K glutamate, 50 mM HEPES pH 7.5, 1 mM EDTA pH 8.0) plus protease inhibitors and stirred slowly for 30′. Cell debris was removed by centrifugation (19,000 r.p.m. in a SS-34 rotor for 1 hr at 4°C) and the supernatant was collected and mixed with 1.5 ml anti-Flag M2 affinity resin (Sigma) equilibrated in 1X FLAG binding buffer with 10% glycerol. The mixture was rotated on an orbital platform in the cold room at 30 rpm for 1 hr. To collect the bound proteins, anti-FLAG resin was pelleted at 1000 X g in 50 ml conical tubes and washed 5 times with 5 ml of FLAG binding buffer followed by centrifugation. After the final wash step, the anti-Flag affinity resin was resuspended in 2 ml of FLAG binding buffer with 10%

glycerol, loaded onto a gravity column and washed twice with 7.5 ml of FLAG binding buffer containing 10% glycerol. Bound protein was eluted with the same buffer containing 0.2 mg/ml 3X FLAG peptide (EZ Biolab, Carmel, Indiana USA). Eluted protein was concentrated to 0.75 ml of 1.5 mg/ml protein and injected onto a 24 ml Superose 12 gel filtration column equilibrated in 2X PBS with 10% glycerol in two separate runs of 0.5 ml and 0.25 ml. Fractions were analyzed on a 7.5% SDS-PAGE gel and MTC-containing fractions were pooled, aliquoted, flash frozen in liquid nitrogen, and stored at −80℃.

## Helicase substrates

For all radiolabeled oligonucleotides, 10 pmol of oligonucleotide was labeled at the 5' terminus with 0.05 mCi [γ-$^{32}$P]-ATP using T4 Polynucleotide Kinase (New England Biolabs) in a 25 µl reaction for 30' at 37℃ according to the manufacturer's instructions. The kinase was heat inactivated for 20' at 80℃. For annealing, 4 pmol of the radiolabeled strand was mixed with 6 pmol of the unlabeled complementary strand, NaCl was added to a final concentration of 200 mM, and the mixture was heated to 90℃ and then cooled to room temperature over a time frame of >1 hr. DNA oligonucleotides used in this study are listed in *Table 1*.

**Table 1.** Oligonucleotides Used in this Study.
All oligonucleotides used in this study were ordered from IDT with the indicated modifications.

| Oligo name | Sequence (5' to 3') | Modification(s) |
|---|---|---|
| 50dupex LAG | TTTTTTTTTTTTTTTTTTTTTTTTTTTTTTTTTTTTGACGCTGCC GAATTCTGGCTTGCTAGGACATTACAGGATCGTTCGGTCTC | None |
| 50duplex LAG dual biotin | TTTTTTTTTTTTTTTTTTTTTTTTTTTTTTTTTTTGACG CTGCCGAA**T**TCTGGC**T**TGCTAGGACATTACAGGATCGTTCGGTCTC | Two biotin-modified thymidine residues in **BOLD** |
| 50duplex LEAD | GAGACCGAACGATCCTGTAATGTCCTAGCAAGCCAGAATTCGGC AGCGTCTTTTTTTTTTTTTTTTTTTTTTTTTTTTTTT*T*T*T*T*T*T | The six dT residues at the 3' end are connected by phosphorothioate bonds (*) |
| 50duplex LAG2 | TTTTTTTTTTTTTTTTTTTTTTTTTTTTTTTTTTTTTTGACGCT GCCGAATTCTGGATTGCTAGGACATTACAGGATCGTTCGGTCTC | None |
| 50duplex LEAD2 dual biotin | GAGACCGAACGATCCTGTAATGTCCTAGCAA**T**CCA GAA**T**TCGGCAGCGTCTTTTTTTTTTTTTTTTTTTTTTTTTTTTTTT*T*T*T*T*T*T | Two biotin-modified thymidine residues in **BOLD**; the six dT residues at the 3' end are connected by phosphorothioate bonds (*) |
| 160mer duplex LEAD | AGAGAGTAGAGTTGAGTTGTGATGTGTAGAGTTGTTGTAGAG AAGAGTTGTGAAGTGTTGAGTAGAGAAGAGAAGAGAAGTGTTGTG ATGTGTTGAGTAGTGTAGAGTTGAGAAGTAGAGATGTGTTGAGATGAGAAGAGTTGTA GTTGAGTTGAAGTGGTTTTTTTTTTTTTTTTTTTTTTTTTTTTTTTTT*T*T*T*T*T | The five dT residues at the 3' end are connected by phosphorothioate bonds (*) |
| 160mer duplex LAG | TTTTTTTTTTTTTTTTTTTTTTTTTTTTTTTTTTTTCCACTTCAAC TCAACTACAACTCTTCTCATCTCAACACATCTCTACTTCTCAACTCTACAC TACTCAACACATCACAACACTTCTCTTCTCTTCTCTACTCAACACTTCACAA CTCTTCTCTACAACAACTCTACACATCACAACTCAACTCTACTCTCT | None |
| Blocked Fork LEAD | ACCGGAGACCGAACGATCCTGTAATGTCCTAGCAA GCCAGAATTCGGCAGCGTCTTTTTTTTTTTTTTTTTTTTTTTTTTTTTTTTT TGAGGAAAGAATGTTGGTGAGGGTTGGGAAGTGGAAGGATGGGCTC GAGAGGTTTTTTTTTTTTTTTTTTTTTTTTTTTTTT*T*T*T*T*T | The five dT residues at the 3' end are connected by phosphorothioate bonds (*) |
| Blocked Fork LAG | TTTTTTTTTTTTTTTTTTTTTTTTTTTTTTTTTTTTTTTTTGACGC TGCCGAA**T**TCTGGC**T**TGCTAGGACATTACAGGATCGTTCG*G*T*C*T*C | Two biotin-modified thymidine residues in **BOLD**; the five dT residues at the 3' end are connected by phosphorothioate bonds (*) |
| Blocked Fork Primer | CCTCTCGAGCCCATCCTTCCACTTCCCAACCCTCACC | None |
| C2 | CCTCTCGAGCCCATCCTTCCACTTCCCAACCCTCACC | None |

DOI: https://doi.org/10.7554/eLife.29118.017

## Helicase assays with forked DNA substrates

For the assays in *Figures 2* and *3C*, the forked DNA was formed using the following two oligos (*Table 1*): 50duplex LEAD and 5'-$^{32}$P-50duplex LAG. For the assays in *Figure 3B*, the forked DNA was formed using unlabeled 160mer duplex LEAD and 5'-$^{32}$P-160mer duplex LAG. Oligos were annealed as described above.

Reactions in *Figure 2A* contained 25 nM CMG and either 0 nM, 25 nM, 50 nM or 100 nM of Mcm10 (as indicated) with 0.5 nM DNA substrate and 1 mM ATP in 40 µl final volume of buffer A (20 mM Tris Acetate pH 7.6, 5 mM DTT, 0.1 mM EDTA, 10 mM MgSO$_4$, 30 mM KCl, 40 µg/ml BSA). Reactions were mixed on ice and started by placing in a water bath at 30° C. 1' after starting the reaction, 25 nM unlabeled 50duplex LAG oligo was added as a trap to prevent re-annealing of unwound radiolabeled DNA. At the indicated times, 12 µl aliquots were removed, stopped with buffer containing 20 mM EDTA and 0.1% SDS (final concentrations), and flash frozen in liquid nitrogen. Frozen reaction products were thawed quickly in water at room temperature and separated on 10% native PAGE minigels in TBE buffer. Gels were washed in distilled water, mounted on Whatman 3 MM paper, wrapped in plastic and exposed to a phosphor screen that was scanned on a Typhoon 9400 laser imager (GE Healthcare). Scanned gels were analyzed using ImageQuant TL v2005 software (e.g. for *Figure 2B and C*). For all quantitations of helicase assays, the small % background of unannealed radiolabeled primer in the 'No CMG' lane was subtracted from the % unwound at each time point.

Reaction conditions in *Figure 3* were similar to those in *Figure 2A* but using 20 nM CMG and 40 nM Mcm10 (where indicated). CMG was mixed with the substrate on ice in the absence of ATP and placed at 30° C for 10' to allow CMG to load onto the substrates without unwinding. To start the reaction, ATP was added with or without Mcm10 (as indicated). 1' after starting the reaction, 50 nM unlabeled lagging strand oligo was added as a trap to prevent re-annealing of unwound radiolabeled DNA. Total reaction volumes were 126 µl, and 11 µl aliquots were stopped at the indicated times after addition of ATP and processed as described for the assays of *Figure 2A*.

## Helicase assays using a dual biotin fork DNA

For the assays in *Figure 5*, the forked DNA was formed by annealing 50 duplex LAG dual biotin and 5'-$^{32}$P-50duplex LEAD (see *Table 1*). The biotinylated dT nucleotides are 13 and 20 bases from the forked junction. For the assays in *Figure 5—figure supplement 1*, the forked DNA was formed by annealing 50 duplex LEAD2 dual biotin and 5'-$^{32}$P-50duplex LAG2 (see *Table 1*). The biotinylated dT nucleotides are 12 and 19 bases from the forked junction. Oligos were annealed as described above. Reaction conditions were similar to those in *Figure 3* except that the final reaction volume was 45 µl and 4 µg/ml streptavidin was added (where indicated) during the 10' CMG pre-incubation. CMG was at 25 nM and Mcm10 or MTC was at 50 nM (final concentrations) when present. In these assays, the trap oligo was 50 nM unlabeled 50duplex LEAD oligo. 12 µl aliquots were removed at the indicated times after addition of ATP, terminated with EDTA/SDS stop buffer, flash frozen and processed as above. For the assays in *Figure 6*, the radiolabel was on the biotinylated strand and, where indicated, free biotin was added at the start of the reaction (after streptavidin pre-incubation) to a final concentration of 1.5 µM to prevent re-binding of any streptavidin displaced by CMG/Mcm10 during the unwinding reaction.

## Mcm10 binding to CMG

To determine if Mcm10 binds to CMG in a stable fashion (*Figure 2D*), we mixed 40 pmol of FLAG-CMG with 120 pmol Mcm10. The mixture was incubated for 15 min on ice and then spun in a microcentrifuge at 15,000 rpm for 10' at 4°C. The volume of the protein solution was adjusted to 150 µl with binding buffer (25 mM Hepes, pH 7.5; 10% glycerol; 0.01% Nonidet P-40; 300 mM NaCl) and mixed with 25 µl anti-FLAG M2 magnetic beads (50% suspension; Sigma-Aldrich, St. Louis, MO). The protein-bead mixture was incubated on ice for 1 hr and then the beads were collected with a magnetic separator and the supernatant (containing unbound proteins) was removed. The beads were washed three times with 250 µl binding buffer and bound proteins were eluted by incubating in 62.5 µl of the same buffer supplemented with 0.2 mg/ml 3X FLAG peptide on ice for 30'. The beads were collected with a magnetic separator and eluted proteins were collected and analyzed in an 8% SDS-polyacrylamide gel stained with Denville Blue.

## Replication assays with 2.8 kb duplex substrate

Leading strand replication experiments in *Figure 4* used a singly primed 2.8 kb forked linear DNA substrate that was previously described (*Georgescu et al., 2014*). The duplex portion of the DNA substrate is linearized pUC19 DNA to which a synthetic fork junction has been ligated to one end of the duplex. The fork is primed for leading strand DNA replication with 5'-$^{32}$P-C2 oligo (*Table 1*). Reactions were 25 µL and contained 30 nM CMG, 10 nM Pol ε, 5 nM RFC, 25 nM PCNA, 600 nM RPA and 1.25 nM linear forked template (final concentrations) in a buffer consisting of 25 mM Tris Acetate pH 7.5, 5% glycerol, 40 µg/ml BSA, 3 mM DTT, 2 mM TCEP, 10 mM magnesium acetate, 50 mM K glutamate, 0.1 mM EDTA, 5 mM ATP, and 120 µM of each dNTP. Replication assays were performed by first incubating CMG (and the indicated amount of Mcm10 and/or MTC, where indicated) with linear forked template for 5' at 30° C, followed by addition of RFC, PCNA, and Pol ε for 4' in the presence of dATP and dCTP to support clamp loading and polymerase binding while preventing 3'−5' exonuclease activity on the primer. Reactions were started by addition of ATP, RPA, and the withheld nucleotides (dGTP and dTTP). The reactions proceeded for the indicated amount of time at 30°C and were stopped with an equal volume of 2X stop solution (40 mM EDTA and 1% SDS). Reaction products were analyzed on 1.3% alkaline agarose gels at 35 V for 17 hr, backed with DE81 paper, and dried by compression. Gels were exposed to a phosphorimager screen and imaged with a Typhoon FLA 9500 (GE Healthcare).

## Stalled replisome assays

For the experiments in *Figure 5* using a stalled replisome, the substrate was made by annealing Blocked Fork LEAD, Blocked Fork LAG and 5'-$^{32}$P-Blocked Fork Primer (*Table 1*). The lagging strand oligo has two biotinylated dT nucleotides that are 13 and 20 bases from the forked junction. To form the lagging strand block, streptavidin (75 nM) was added to the substrate (1.25 nM) and incubated at 30°C for 5' in buffer consisting of 20 mM Tris Acetate, 4% glycerol, 0.1 mM EDTA, 5 mM DTT, 40 µg/mL BSA and 10 mM MgSO$_4$ (all amounts are the final concentration in the complete reaction). Next, 30 nM CMG was added along with 0.5 mM ATP for 3' to allow CMG to translocate to the block and then replication was initiated by adding 5 nM RFC, 20 nM PCNA, 20 nM Pol ε, 5 mM ATP, and 30 nM MTC (when present) along with adding 115 µM of each dNTP. After a further 2' incubation, Mcm10 was either added at 60 nM or omitted. Aliquots of each reaction were collected 1, 3, and 10 min later and quenched with an equal volume of Stop Buffer containing 78% formamide, 8 mM EDTA, and 1% SDS. Samples were boiled and then analyzed by PAGE in a 10% Urea gel. Gels were washed in distilled water, mounted on Whatman 3 MM paper, wrapped in plastic and exposed to a storage phosphor screen that was scanned on a Typhoon 9400 laser imager (GE Healthcare).

# Additional information

### Funding

| Funder | Grant reference number | Author |
|---|---|---|
| Howard Hughes Medical Institute | | Lance D Langston<br>Ryan Mayle<br>Olga Yurieva<br>Roxana E Georgescu<br>Mike E O'Donnell |
| National Institutes of Health | GM38839 | Lance D Langston<br>Ryan Mayle<br>Grant D Schauer<br>Olga Yurieva<br>Daniel Zhang<br>Nina Y Yao<br>Roxana E Georgescu<br>Mike E O'Donnell |

The funders had no role in study design, data collection and interpretation, or the decision to submit the work for publication.

## Author contributions
Lance D Langston, Conceptualization, Resources, Formal analysis, Validation, Investigation, Writing—original draft, Writing—review and editing; Ryan Mayle, Grant D Schauer, Roxana E Georgescu, Conceptualization, Investigation; Olga Yurieva, Resources; Daniel Zhang, Resources, Investigation; Nina Y Yao, Investigation; Mike E O'Donnell, Conceptualization, Supervision, Funding acquisition, Visualization, Writing—original draft, Project administration, Writing—review and editing

## Author ORCIDs
Lance D Langston ⃝ http://orcid.org/0000-0002-2736-9284
Roxana E Georgescu ⃝ http://orcid.org/0000-0002-1882-2358
Mike E O'Donnell ⃝ http://orcid.org/0000-0001-9002-4214

## Decision letter and Author response
Decision letter https://doi.org/10.7554/eLife.29118.019
Author response https://doi.org/10.7554/eLife.29118.020

# Additional files

## Supplementary files
• Transparent reporting form
DOI: https://doi.org/10.7554/eLife.29118.018

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
