## [Decision Letter]

Thank you for submitting your article "Mcm10 functions to isomerize CMG-DNA for replisome bypass of DNA blocks" for consideration by *eLife*. Your article has been reviewed by three peer reviewers, one of whom is a member of our Board of Reviewing Editors, and the evaluation has been overseen by Philip Cole as the Senior Editor. The reviewers have opted to remain anonymous.

The reviewers have discussed the reviews with one another and the Reviewing Editor has drafted this decision to help you prepare a revised submission.

The manuscript from Langston and O'Donnell investigates the role of MCM10 in eukaryotic helicase function. They show convincingly that a stable complex forms between MCM10 and the CMG, and that the MCM10/CMG complex shows a dramatic improvement in helicase activity over the CMG alone. Although these are significant findings in their own right, they claim that MCM10's primary function is not to improve CMG action per se (as MCM10's effect on the CMG is eclipsed by that of the MTC complex), but to fundamentally alter the mechanism of translocation, isomerizing the CMG between a double-stranded to single-stranded encirclement mechanism for the purpose of bypassing blocks. All known hexameric DNA helicases translocate on one strand and exclude the other strand of a DNA fork. Langston, O'Donnell and colleagues previously observed that the eukaryotic replicative helicase, CMG, cannot bypass a roadblock on either the translocation strand (leading strand template) or the sterically excluded lagging strand. The structure of a fork-engaged CMG was used to explain this observation: duplex DNA is encircled by a B domain crown and the fork nexus is contacted by the C domain "OB hairpins", which guard the single-stranded DNA interacting ATPase ring pore. Given this mode of binding, a roadblock on either strand of the duplex DNA would therefore be impaired from physically reaching C domain hairpins. Langston and colleagues now reconstitute a CMG-Mcm10 complex and find that Mcm10 makes the CMG a more vigorous helicase, capable of bypassing a roadblock on the lagging strand. The authors suggest that Mcm10 induces an isomerization of the CMG that might release the duplex DNA portion of a fork from the B-domain embedding. This event would occur upon initiation of DNA replication and when the replisome needs to bypass a roadblock on the DNA.

Although MCM10 clearly improves the ability of the CMG to bypass a lagging strand block, their conclusion that this is due to CMG/DNA isomerization may be premature. A simpler explanation is that the MCM10-dependent improvement in helicase activity provides the force needed to drive the CMG through the block. Notably, this is consistent with their previous finding that the CMG alone (in the absence of MCM10) can dislodge a block potentially mediated by a single streptavidin from either strand. Why, then, should not a motor with dramatically improved activity (30-fold!) be able to bypass two such potential road-blocks? Additionally, these conclusions are based on the premise that the CMG does, in fact, encircle two strands. However, the background biochemical work on which this manuscript is based (Langston and ODonnell 2017 *eLife*) came to this conclusion by comparing the CMG's ability to bypass tandem biotin-streptavidin blocks on leading and lagging strand templates, where they find both able to block progression. This result by itself would be consistent with this conclusion but provides no direct evidence for the model. Other models include an outside path for one strand that must interact with the outer surface. Further and perhaps more important beyond model building, for this work and previous studies from the group, they fail to mention in their paper that the two bulky adducts on the lagging strand are placed at half the distance as those on the leading strand. Although their recent EM structure suggests that DNA forks engage further within the protein than initially anticipated, the lagging strand is still, nonetheless, excluded, and double-stranded DNA never enters the central channel -how deep one must go to be in the "central channel" is, of course, perhaps semantics.

Summary:

All three reviewers thought that the issues raised above regarding the isomerization model were not convincing and needed to be either presented with more direct structural data or presented as a speculation amongst other possibilities. A revised manuscript should address the essential and minor issues enumerated below.

Essential revisions:

1) Their use of non-equivalent leading and lagging strand dual biotin-streptavidin roadblocks is misleading and correctly matched oligos should be ordered and the experiments repeated (including the basic finding that lagging strand adducts block translocation equally as well as leading strand adducts)

2) Their title overstates their findings – they have not shown isomerization, only that MCM10 improves the helicase function of the CMG. Either the idea is presented as a speculation along side other ideas or direct data supporting the point be presented as new information

3) Their work suggests an MCM10-dependent switching of the CMG between a double-stranded to single-stranded mechanism upon encountering a roadblock. But given the performance advantage of MCM10 on the CMG and, as the authors state, the persistence of MCM10 at forks, why should the CMG not always occupy this functional state, i.e. steric exclusion without ever occupying the (intermediate?) state they observe by EM. This should be discussed as a speculation. On the other hand direct biophysical or structural data provided.

4) Figure 1 cartoon is a bit misleading. The MCM should be depicted as a 3 tier ring: from the bottom, ATPase, A-C tier and smaller B tier. Duplex DNA is engaged by B-tier only. At the moment it appears that fork nexus is found at the ATPase/A-C tier interface, which is not the case.

5) Figure 2, it would be good to include a control where Mcm10 is incubated with a fork in the absence of CMG. Also, one caveat of this experiment is that Mcm10 is a single-stranded DNA binding protein and the stimulation of the DNA unwinding activity might simply be due to single-stranded DNA trapping (effectively the same role of a DNA capture strand). It would be good to test whether another single-stranded DNA binding protein (e.g. SSB) has a similar effect on DNA unwinding by the CMG. Also, what is the supershifted band visible e.g. in lanes 5, 8 and 11? It would be good to acknowledge this band, since it is present in many unwinding gels throughout the paper.

6) Figure 2. GINS subunits are notoriously difficult to stain. It is a pity that in the CMG-Mcm10 gel appears to be cut so that the GINS subunits are not visible. From looking at this gel alone one cannot rule out the possibility that Mcm10 and GINS compete for the same binding site on the MCM. In Figure 2—figure supplement 1 the MonoQ profile is shown where Sld5 is detected, however not Psf1-3. It is remarkable that the CMG-Mcm10 complex is stable at 300mM KCl concentration (indeed convincingly shown in the high salt FLAG pulldown experiment), however the Q column experiment alone does not rule out the possibility that Sld5 and MCM-Cdc45-Mcm10 simply elute at the same salt concentration from an ion exchange column. Could the authors produce a gel where all of the CMG components are seen together with Mcm10, after FLAG elution?

7) Figure 3. The observation that CMG plus Mcm10 is much better at unwinding longer stretches of duplex DNA compared to CMG alone is interesting. Why do the authors think that a fork with 160bp duplex is a better substrate than the 50bp duplex for CMG-Mcm10?

8) Figure 5—figure supplement 1 lanes 11-13. CMG-Mcm10 can clearly bypass a leading strand roadblock, however the authors do not comment on this remarkable observation. Is SA kicked off of the DNA or is there a more complicated leading strand roadblock bypass mechanism to be uncovered? Using DNA-Protein crosslink (methyltransferase?) as a roadblock would address this issue. If SA is kicked off of the leading strand template, this might simply mean that the bypass effect observed here is due to a stimulation of the ATPase activity (CMG-Mcm10 powers through the roadblock). Could the authors address this point? Does Mcm10 stimulate ATP hydrolysis by MCM to a different extent compared to e.g. MTC, which the authors already use as an informative control in this study?

From these experiments it is not clear that streptavidin remains bound to the substrate after CMG passage, as invoked by the bypass mechanism proposed by the authors. This can be addressed by placing the 32P-label on the modified strand of the DNA substrate to observe the generation of single-stranded streptavidin-DNA products, and by performing the reactions in the presence of excess free biotin to trap displaced streptavidin. This seems particularly relevant, as the experiment in Figure 5—figure supplement 1 detects significant Mcm10-dependent bypass of blocks even on the leading strand – how do the authors explain this latter observation?

9) The four GINS subunits are not visible in the gel pictures of the isolated CMG-Mcm10 complexes in Figure 2 and Figure 2—figure supplement 1. Evidence for their stoichiometric presence in the complex needs to be presented.

---

## [Author Response]

Essential revisions:1) Their use of non-equivalent leading and lagging strand dual biotin-streptavidin roadblocks is misleading and correctly matched oligos should be ordered and the experiments repeated (including the basic finding that lagging strand adducts block translocation equally as well as leading strand adducts)

We appreciate the reviewer’s comment and have repeated the experiments as requested. We now space the two biotinylated T residues by 6 bp on each strand (this is the spacing used on the lagging strand of the first submission). The results are presented in Figure 5 and Figure 6 (for lagging blocks) and Figure 5—figure supplement 1 and Figure 6 (for leading blocks). Figure 5 has been expanded by taking one figure from the supplement and adding it to the main figure (now parts D and E). The closer spaced blocks on the leading strand are more inhibitory than the 14bp spacing in the earlier manuscript, and therefore the more exact comparison was important to the outcome, and we thank the reviewers for this suggestion.

2) Their title overstates their findings – they have not shown isomerization, only that MCM10 improves the helicase function of the CMG. Either the idea is presented as a speculation along side other ideas or direct data supporting the point be presented as new information

We thank the reviewers and have added data demonstrating that the lagging strand blocks are circumvented without displacement (i.e. using an excess biotin trap), while leading blocks are displaced. The data are presented in Figure 6 and discussed in the text (see detailed response to comment #8). We also discuss the ring opening type of isomerization that may occur, in addition to the possible steric exclusion isomerization:

“Two alternative isomerization reactions that would also facilitate lagging strand block bypass without displacement include one in which Mcm10 cracks open the N-tier of the Mcm2-7 ring of CMG, bypasses the block, and recloses the N-tier. Similarly, Mcm10 might enable opening of both the N- and C-tiers, either simultaneously or in succession, to bypass the blocks. These alternative isomerization reactions are described in detail in Figure 8 of (Langston et al., 2017)”

Also, considering that only lagging blocks are bypassed without displacement, we have changed the title to: “Mcm10 promotes rapid isomerization of CMG-DNA for bypass of lagging strand blocks”.

3) Their work suggests an MCM10-dependent switching of the CMG between a double-stranded to single-stranded mechanism upon encountering a roadblock. But given the performance advantage of MCM10 on the CMG and, as the authors state, the persistence of MCM10 at forks, why should the CMG not always occupy this functional state, i.e. steric exclusion without ever occupying the (intermediate?) state they observe by EM. This should be discussed as a speculation. On the other hand direct biophysical or structural data provided.

We appreciate the reviewer’s comments, and agree that a stable CMG-Mcm10 complex is suggestive that Mcm10 always travels with the replisome. But the available cellular studies that address whether Mcm10 travels with the replisome are not settled in the literature, with reports and data going in both directions. Since we have not shown the persistence of Mcm10 (i.e. k_off_ rate) with CMG at a fork, we discuss the possibility that CMG might travel with the fork all the time in the Discussion, as requested, but we also note that we cannot rigorously make that conclusion from the data of the current report.

4) Figure 1 cartoon is a bit misleading. The MCM should be depicted as a 3 tier ring: from the bottom, ATPase, A-C tier and smaller B tier. Duplex DNA is engaged by B-tier only. At the moment it appears that fork nexus is found at the ATPase/A-C tier interface, which is not the case.

We thank the reviewers for pointing this out. We did not intend to draw the unwinding point as far down as we did, and this was an artistic flaw of the original figure. We have now redrawn the figure to show a much smaller amount of dsDNA entering the CMG. We did not redraw the cartoon to have 3 tiers because describing the A, B, C domains of the N-tier is beyond the scope of the Introduction. We have dealt with these aspects in the structural study and in a few other reviews that are submitted and underway.

5) Figure 2, it would be good to include a control where Mcm10 is incubated with a fork in the absence of CMG. Also, one caveat of this experiment is that Mcm10 is a single-stranded DNA binding protein and the stimulation of the DNA unwinding activity might simply be due to single-stranded DNA trapping (effectively the same role of a DNA capture strand). It would be good to test whether another single-stranded DNA binding protein (e.g. SSB) has a similar effect on DNA unwinding by the CMG. Also, what is the supershifted band visible e.g. in lanes 5, 8 and 11? It would be good to acknowledge this band, since it is present in many unwinding gels throughout the paper.

We appreciate these several comments and address them below.

– Mcm10 alone displays no helicase activity and we have now added this data as Figure 2—figure supplement 1.

– We long ago performed the experiments that ask whether RPA or SSB might stimulate CMG unwinding prior to submitting the original manuscript, and have now included this data as a Figure 2—figure supplement 2. We should note that SSB/RPA can’t be added to reactions before CMG as it prevents CMG loading (as we published earlier). Thus, we let the CMG helicase reaction proceed for 10 min, and then add either Mcm10, RPA or SSB. The result shows that Mcm10 provides strong stimulation of the CMG helicase while neither RPA nor SSB affect CMG activity.

– The supershifted band in lanes 5,8 and 11 has now been acknowledged and explained in the text. The second half of the first paragraph in the Results section describes these 3 points and reads as follows:

“Control reactions with Mcm10 alone, lacking CMG, show no unwinding activity (Figure 2—figure supplement 1). Reactions with Mcm10 sometimes give a supershift of some of the substrate (e.g. lanes 5, 8, 11, 12 in Figure 2), which we interpret as a gel shift of Mcm10 that remains bound to DNA, as Mcm10 is a known DNA binding protein (Du et al., 2012; Robertson et al., 2008; Warren et al., 2008). To determine if other DNA binding proteins such as RPA or *E. coli* SSB stimulate CMG, we performed experiments in which CMG unwinding was initiated and then either Mcm10, RPA or SSB was added. The results show that the stimulation of CMG is specific to Mcm10 and that RPA and SSB do not substitute for Mcm10 (Figure 2—figure supplement 2).”

6) Figure 2. GINS subunits are notoriously difficult to stain. It is a pity that in the CMG-Mcm10 gel appears to be cut so that the GINS subunits are not visible. From looking at this gel alone one cannot rule out the possibility that Mcm10 and GINS compete for the same binding site on the MCM. In Figure 2—figure supplement 1 the MonoQ profile is shown where Sld5 is detected, however not Psf1-3. It is remarkable that the CMG-Mcm10 complex is stable at 300mM KCl concentration (indeed convincingly shown in the high salt FLAG pulldown experiment), however the Q column experiment alone does not rule out the possibility that Sld5 and MCM-Cdc45-Mcm10 simply elute at the same salt concentration from an ion exchange column. Could the authors produce a gel where all of the CMG components are seen together with Mcm10, after FLAG elution?

We appreciate the reviewer’s concerns and certainly agree that the GINS are notoriously difficult to stain. We have rerun the gels, but we should also point out that while this project was being performed, Mcm10 was demonstrated in the Bell lab to preferentially bind CMG, containing GINS, Cdc45 and Mcms (Lõoke, et al., 2017). However, we have re-run FLAG and MonoQ experiments and re-run gels to the best we could, and have added those gels and the following text to the manuscript. We also note that Steve Bell’s recent paper on Mcm10 (cited herein) demonstrates that Mcm10 stabilizes the CMG complex.

– We have performed a repeat of the FLAG purification of CMG-Mcm10 and run the gel such that the GINS subunits can be seen (Figure 2—figure supplement 3). Densitometric analysis, described in the legend to Figure 2—figure supplement 3, reads:

“To determine the stoichiometry of GINS in the CMG-Mcm10 complex, a Flag purification of CMG-Mcm10 was analyzed in a 10% PAGE gel followed by densitometric analysis. The result indicates the following stoichiometry: 1.0 Mcm2-7, 1.2 Mcm10, 0.87 Sld5, and 2.6 Psf1+Psf2 (they co-migrate for an average of 1.3 each). The Psf3 runs at the dye front and could not be quantitated. FLAG columns capture some free Cdc45 and thus overestimate the Cdc45-FLAG subunit; MonoQ analysis removes free Cdc45 (see Figure 2—figure supplement 4).”

– We have also re-run the MonoQ purification of CMG-Mcm10 complex in the PAGE such that the GINS subunits are visible (Figure 2—figure supplement 4). The densitometric scan information in the legend reads as follows:

“A densitometric scan of the gel showed a molar ratio of 1.0 Mcm2-7: 1.59 Cdc45: 2.0 Mcm10: 1.4 Sld5: Psf1 and Psf2 at 1.05 each (Psf1/2 co-migrate, and together give a stoichiometry of 1.0:2.1 for Mcm2-7: Psf1+Psf2); Psf3 runs at the dye front.”

– In the body of the text we have added the following:

“The GINS subunits of the CMG-Mcm10 complex ran off the gel, but a previous study demonstrated that Mcm10 preferentially associates with the entire CMG complex (Lõoke et al., 2017). This is supported by densitometry analysis of gels showing GINS subunits in both FLAG and MonoQ purified CMG-Mcm10 complexes (Figure 2—figure supplement 3 and Figure 2—figure supplement 4).”

We note to reviewers (not in the text), that scans of Commassie stained bands are only approximations because different proteins can take up different amounts of stain, but that the results overall indicate that Mcm10 binds CMG without displacing the GINS and Cdc45 accessory factors.

We have also recently performed cross-linking/mass spectrometry on CMG-Mcm10 complex, and find that Mcm10 runs across the surface of two of the GINS subunits and the Cdc45 subunit, and wraps around 3 of the six Mcm2-7 subunits. The CX-MS results are not included here, because they are part of our study on the structure of CMG-Mcm10, but they confirm that GINS and Mcm10 are associated with CMG, and that Mcm10 binds a lot of the CMG subunits, which probably accounts for Steve Bell’s observation in his recent study that Mcm10 preferentially binds the complete CMG, and facilitates formation of the CMG complex (Lõoke, et al., 2017).

7) Figure 3. The observation that CMG plus Mcm10 is much better at unwinding longer stretches of duplex DNA compared to CMG alone is interesting. Why do the authors think that a fork with 160bp duplex is a better substrate than the 50bp duplex for CMG-Mcm10?

We apologize for the misconception in our text. We did not mean to imply that the fork with a 160bp duplex is a better substrate than a fork with a 50bp duplex. We meant to say that these appear to be equal substrates for CMG-Mcm10. We have clarified this point in the text of the revised manuscript. We have added the following statement to the Results:

“The final extent of unwinding is similar for both substrates, so the delay reflects a difference in time of unwinding the two DNAs due to their difference in length rather than a difference in helicase activity on the two substrates.”

8) Figure 5—figure supplement 1 lanes 11-13. CMG-Mcm10 can clearly bypass a leading strand roadblock, however the authors do not comment on this remarkable observation. Is SA kicked off of the DNA or is there a more complicated leading strand roadblock bypass mechanism to be uncovered? Using DNA-Protein crosslink (methyltransferase?) as a roadblock would address this issue. If SA is kicked off of the leading strand template, this might simply mean that the bypass effect observed here is due to a stimulation of the ATPase activity (CMG-Mcm10 powers through the roadblock). Could the authors address this point? Does Mcm10 stimulate ATP hydrolysis by MCM to a different extent compared to e.g. MTC, which the authors already use as an informative control in this study?From these experiments it is not clear that streptavidin remains bound to the substrate after CMG passage, as invoked by the bypass mechanism proposed by the authors. This can be addressed by placing the 32P-label on the modified strand of the DNA substrate to observe the generation of single-stranded streptavidin-DNA products, and by performing the reactions in the presence of excess free biotin to trap displaced streptavidin. This seems particularly relevant, as the experiment in Figure 5—figure supplement 1 detects significant Mcm10-dependent bypass of blocks even on the leading strand – how do the authors explain this latter observation?

We appreciate this comment, and in fact did the excess biotin trap experiment before submission for the lagging strand streptavidin blocks. On hindsight we should have included that data in the original submission. In the revised manuscript, we perform both leading and lagging experiments using a biotin trap (and 32P-label on the biotinylated strand), and using the forked DNAs with equal 6bp spacing between the two biotin-streptavidins for both leading and lagging strands. The results show that Mcm10 enables bypass of the lagging strand blocks without displacing the streptavidin blocks, and that the leading strand streptavidins are in fact displaced from DNA (there was less efficient leading strand read through with the 6 bp spacing compared to the earlier 14 bp spacing). These data are now presented in a newly added Figure 6 and supplement to Figure 6, which shows the effectiveness of the biotin trap. The modified manuscript reads as follows:

“Lagging strand blocks are bypassed while leading strand blocks are displaced. […]Hence, Mcm10 appears to provide CMG with more translocation force than CMG alone, consistent with the increase in processivity of CMG in the presence of Mcm10 (Figure 3).”

9) The four GINS subunits are not visible in the gel pictures of the isolated CMG-Mcm10 complexes in Figure 2 and Figure 2—figure supplement 1. Evidence for their stoichiometric presence in the complex needs to be presented.

We now show supplemental figures of PAGE gels showing and quantitating GINS for both FLAG and MonoQ purified CMG-Mcm10 (Figure 2—figure supplement 3 and Figure 2—figure supplement 4). However, the fact that GINS are present in CMG-Mcm10 has been recently published. The text reads:

“The GINS subunits of the CMG-Mcm10 complex ran off the gel, but a previous study demonstrated that Mcm10 preferentially associates with the entire CMG complex (Lõoke, et al., 2017). This is supported by densitometry analysis of gels showing GINS subunits in both the FLAG and MonoQ purified CMG-Mcm10 complexes (Figure 2—figure supplement 3 and Figure 2—figure supplement 4).”

Please see also the response to comment #6, which overlaps with this comment.